# Think out of the "Box": Generically-Constrained Asynchronous Composite Optimization and Hedging

**Pooria Joulani**[*]
DeepMind, UK
pjoulani@google.com

**András György**
DeepMind, UK
agyorgy@google.com

**Csaba Szepesvári**
DeepMind, UK
szepi@google.com

## Abstract

We present two new algorithms, ASYNCADA and HEDGEHOG, for asynchronous sparse online and stochastic optimization. ASYNCADA is, to our knowledge, the first asynchronous stochastic optimization algorithm with finite-time data-dependent convergence guarantees for generic convex constraints. In addition, ASYNCADA: (a) allows for proximal (i.e., composite-objective) updates and adaptive step-sizes; (b) enjoys any-time convergence guarantees without requiring an exact global clock; and (c) when the data is sufficiently sparse, its convergence rate for (non-)smooth, (non-)strongly-convex, and even a limited class of non-convex objectives matches the corresponding serial rate, implying a theoretical "linear speed-up". The second algorithm, HEDGEHOG, is an asynchronous parallel version of the Exponentiated Gradient (EG) algorithm for optimization over the probability simplex (a.k.a. Hedge in online learning), and, to our knowledge, the first asynchronous algorithm enjoying linear speed-ups under sparsity with non-SGD-style updates. Unlike previous work, ASYNCADA and HEDGEHOG and their convergence and speed-up analyses are not limited to individual coordinate-wise (i.e., "box-shaped") constraints or smooth and strongly-convex objectives. Underlying both results is a generic analysis framework that is of independent interest, and further applicable to distributed and delayed feedback optimization.

## 1 Introduction

Many modern machine learning methods are based on iteratively optimizing a regularized objective. Given a convex, non-empty set of feasible model parameters $\mathcal{X} \subset \mathbb{R}^d$, a differentiable *loss* function $f : \mathbb{R}^d \to \mathbb{R}$, and a convex (possibly non-differentiable) *regularizer* function $\phi : \mathbb{R}^d \to \mathbb{R}$, these methods seek the parameter vector $x^* \in \mathcal{X}$ that minimizes $f + \phi$ (assuming a minimizer exists):

$$x^* = \arg\min_{x \in \mathcal{X}} f(x) + \phi(x) . \tag{1}$$

In particular, empirical risk minimization (ERM) methods such as (regularized) least-squares, logistic regression, LASSO, and support vector machines solve optimization problems of the form (1). In these cases, $f(x) = \frac{1}{m} \sum_{i=1}^{m} F(x, \xi_i)$ is the average of the loss $F(x, \xi_i)$ of the model parameter $x$ on the given training data $\xi_1, \xi_2, \ldots, \xi_m$ and $\phi(x)$ is a norm (or a combination of norms) on $\mathbb{R}^d$ (e.g., $F(x, \xi) = \log(1 + \exp(x^\top \xi))$ and $\phi(x) = \frac{1}{2}\|x\|_2^2$ in linear logistic regression [13]).

To bring the power of modern parallel computing architectures to such optimization problems, several papers in the past decade have studied parallel variants of the stochastic optimization algorithms applied to these problems. Here one of the main questions is to quantify the cost of parallelization, that is, how much extra work is needed by a parallel algorithm to achieve the same accuracy as its serial variant. Ideally, a parallel algorithm is required to do no more work than the serial version, but

---

[*]Work partially done when the author was at the University of Alberta, Edmonton, AB, Canada.

this is very hard to achieve in our case. Instead, a somewhat weaker goal is to ensure that the price of parallelism is at most a constant factor: that is, the parallel variant needs at most constant-times more updates (or work). In other words, using $\tau$ parallel process requires a wall-clock running time that is only $O(1/\tau)$-times that of the serial variant. In this case we say that the parallel algorithm achieves a *linear speed-up*. Of particular interest are asynchronous lock-free algorithms, where Recht et al. [30] demostrated first that linear speed-ups are possible: They showed that if $\tau$ processes run stochastic gradient descent (SGD) and apply their updates to the same shared iterate without locking, then the overall algorithm (called Hogwild!) converges after the same amount of work as serial SGD, up to a multiplicative factor that increases with the number of concurrent processes and decreases with the sparsity of the problem. Thus, if the problem is sparse enough, this penalty can be considered a constant, and the algorithm achieves linear speed-up. Several follow-up work (see e.g., [20, 18, 17, 27, 24, 10, 29, 7, 11, 4, 2, 3, 19, 31, 33, 32, 35, 36, 12, 6, 28] and the references therein) have demonstrated linear speed-ups for methods based on (block-)coordinate descent (BCD), as well as other variants of SGD such as SVRG [15], SAGA [8], ADAGRAD [22, 9], and SGD with a time-decaying step-size. Despite the great advances, however, several problems remain open.[2]

First, the existing convergence guarantees concern SGD when the constraint set $\mathcal{X}$ is box-shaped, that is, a Cartesian product of (block-)coordinatewise constraints $\mathcal{X} = \times_{i=1}^{d} \mathcal{X}_i$. This leaves it unclear whether existing techniques apply to stochastic optimization algorithms that operate on non-box-shaped constraints (e.g., on the $\ell_2$ ball), or algorithms that use a non-Euclidean regularizer, such as the exponentiated gradient (EG) algorithm used on the probability simplex (see, e.g., [34, 14]).

Second, with the exception of the works of Duchi et al. [10] and Pan et al. [26] (which still require box-shaped constraints), and De Sa et al. [7] (which only bounds the probability of "failure", i.e., of producing no iterates in the $\epsilon$-ball around $x^*$), the existing analyses demonstrating linear speed-ups are limited to strongly-convex (or Polyak-Łojasiewicz) objectives. Thus, so far it has remained unclear whether a similar speed-up analysis is possible if the objective is simply convex or smooth [20], or if we are in the closely-related online-learning setting with the objective changing over time.

Third, with the exception of the work of Pedregosa et al. [27] (which still requires box-shaped constraints, block-separable $\phi$ and strongly-convex $f$), the existing analyses do not take advantage of the structure of problem (1). In particular, when $\phi$ is "simple to optimize" over $\mathcal{X}$ (formally defined as having access to a proximal operator oracle, as we make precise in what follows), serial algorithms such as Proximal-SGD take advantage of this property to achieve considerably faster convergence rates. Asynchronous variants of the Proximal-SGD algorithm with such faster rates have so far been unavailable for non-strongly-convex objectives and non-box constraints.

## 1.1 Contributions

In this paper we address the aforementioned problems and present algorithms that are applicable to general convex constraint sets, not just box-shaped $\mathcal{X}$, but still achieve linear speed-ups (under sparsity) for non-smooth and non-strongly-convex (as well as smooth or strongly convex) objectives, and even for a specific class of non-convex problems. This is achieved through our new asynchronous optimization algorithm, ASYNCADA, which generalizes the ASYNC-ADAGRAD (and ASYNC-DA) algorithm of Duchi et al. [10] to proximal updates and its data-dependent bound to arbitrary constraint sets. Instantiations of ASYNCADA under different settings are given in Table 1. Indeed, the results are obtained by a more general analysis framework, built on the work of Duchi et al. [10], that yields data-dependent convergence guarantees for a generic class of adaptive, composite-objective online optimization algorithms undergoing perturbations to their "state". We further use this framework to derive the first asynchronous online and stochastic optimization algorithm with non-box constraints that uses non-Euclidean regularizers. In particular, we present and analyze HEDGEHOG, the parallel asynchronous variant of the EG algorithm, also known as Hedge in online linear optimization [34, 14],

| Algorithm | $\mathcal{X}$ | Nonsmooth | Smooth $f$ | Strongly-convex | Smooth $f$ + Strongly-convex |
|---|---|---|---|---|---|
| SGD (DA) | $\mathbb{R}^d$ | [10, 26] ✓ | [26] ✓ | [26] ✓ | [30, 7, 20, 17, 24, 26] ✓ |
| SGD (MD) | □ | [10, 26] | [26] | [26] | [30, 7, 20, 17, 24, 26] |
| DA | ◯ | ✓ | ✓ | ✓ | ✓ |
| AG / DA | □ | [10, 26] ✓ | [26] ✓ | [26] ✓ | [26] ✓ |
| AG / DA | ◯ | ✓ | ✓ | ✓ | ✓ |
| Prox-MD | □ | - | - | - | [27] |
| Prox-DA | ◯ | ✓ | ✓ | ✓ | ✓ |
| Prox-AG | ◯ | ✓ | ✓ | ✓ | ✓ |
| Hedge/EG | △ | ✓ | ✓ | ✓ | ✓ |

Table 1: (Star-)convex optimization settings under which sufficient sparsity results in linear speed-up. Previous work are cited under the settings they address. A ✓ indicates a setting covered by the results in this paper. The symbols □, △, and ◯ indicate, respectively, the case when the constraint set is box-shaped, the probability simplex, or any convex constraint set with a projection oracle. AG, DA, and MD stand, respectively, for ADAGRAD, Dual-Averaging, and Mirror Descent, while Prox-AG, Prox-DA, and Prox-MD denote their proximal variants (using the proximal operator of $\phi$).

and show that it enjoys similar parallel speed-up regimes as ASYNCADA. The results are derived for the more general setting of noisy online optimization, and the generic framework is of independent interest, in particular in the related settings of distributed and delayed-feedback learning.

The rest of the paper is organized as follows: The optimization problem and its solution with serial algorithms are described in Section 2 and Section 3, respectively. The generic perturbed-iterate framework is given in Section 4. Our main algorithms, ASYNCADA and HEDGEHOG are presented and analyzed in Section 5 and Section 6, respectively. Conclusions are drawn and some open problems are discussed in Section 7, while omitted technical details are given in the appendices.

## 1.2 Notation and definitions

We use $[n]$ to denote the set $\{1, 2, \ldots, n\}$, $\mathbb{I}\{\mathcal{E}\}$ for the indicator of an event $\mathcal{E}$, and $\sigma(\mathcal{H})$ to denote the sigma-field generated by a set $\mathcal{H}$ of random variables. The $j$-th coordinate of a vector $a \in \mathbb{R}^d$ is denoted $a^{(j)}$. For $\alpha \in \mathbb{R}^d$ with positive entries, $\|\cdot\|_\alpha$ denotes the $\alpha$-weighted Euclidean norm, given by $\|x\|_\alpha^2 = \frac{1}{2} \sum_{j=1}^d \alpha^{(j)} \left(x^{(j)}\right)^2$, and $\|\cdot\|_{\alpha,*}$ its dual. We use $(a_t)_{t=i}^j$ to denote a sequence $a_i, a_{i+1}, \ldots, a_j$ and define $a_{i:j} := \sum_{t=i}^j a_t$, with $a_{i:j} := 0$ if $i > j$. Given a differentiable function $h : \mathbb{R}^d \to \mathbb{R}$, the *Bregman divergence* of $y \in \mathbb{R}^d$ from $x \in \mathbb{R}^d$ with respect to (w.r.t.) $h$ is given by $\mathcal{B}_h(y, x) := h(y) - h(x) - \langle \nabla h(x), y - x \rangle$. It can be shown that a differentiable function is convex if and only if $\mathcal{B}_h(x, y) \geq 0$ for all $x, y \in \mathbb{R}^d$. The function $h : \mathbb{R}^d \to \mathbb{R}$ is $\mu$-*strongly convex* w.r.t. a norm $\|\cdot\|$ on $\mathbb{R}^d$ if and only if for all $x, y \in \mathbb{R}^d$ $\mathcal{B}_h(x, y) \geq \frac{\mu}{2}\|x - y\|^2$, and *smooth* w.r.t. a norm $\|\cdot\|$ if and only if for all $x, y \in \mathbb{R}^d$, $|\mathcal{B}_h(x, y)| \leq \frac{1}{2}\|x - y\|^2$. A differentiable function $f$ is *star-convex* if and only if there exists a global minimizer $x^*$ of $f$ such that for all $x \in \mathbb{R}^d$, $\mathcal{B}_f(x^*, x) \geq 0$.

## 2 Problem setting: noisy online optimization

We consider a generic iterative optimization setting that enables us to study both online learning and stochastic composite optimization. The problem is defined by a (known) constraint set $\mathcal{X}$ and a (known) convex (possibly non-differentiable) function $\phi$, as well as differentiable functions $f_1, f_2, \ldots$ about which an algorithm learns iteratively. At each iteration $t = 1, 2, \ldots$, the algorithm picks an iterate $x_t \in \mathcal{X}$, and observes an unbiased estimate $g_t \in \mathbb{R}^d$ of the gradient $\nabla f_t(x_t)$, that is, $\mathbb{E}\{g_t|x_t\} = \nabla f_t(x_t)$. The goal is to minimize the composite-objective online regret after $T$ iterations, given by

$$R_T^{(f+\phi)} = \sum_{t=1}^T \left( f_t(x_t) + \phi(x_t) - f_t(x_T^*) - \phi(x_T^*) \right),$$

where $x_T^* = \arg\min_{x \in \mathcal{X}} \left\{ \sum_{t=1}^{T} (f_t(x) + \phi(x)) \right\}$. In the absence of noise (i.e., when $g_t = \nabla f_t(x_t)$), this reduces to the (composite-objective) online (convex) optimization setting [34, 14].

**Stochastic optimization, online regret, and iterate averaging.** If $f_t = f$ for all $t = 1, 2, \ldots$, we recover the stochastic optimization setting, with the algorithm aiming to minimize the composite objective $f + \phi$ over $\mathcal{X}$ while receiving noisy estimates of $\nabla f$ at points $(x_t)_{t=1}^{T}$. The algorithm's online regret can then be used to control the optimization risk: Since $f_t \equiv f$, we have $x_T^* = x^* = \arg\min_{x \in \mathcal{X}} \{ f(x) + \phi(x) \}$, and by Jensen's inequality, if $f$ is convex and $\bar{x}_T = \frac{1}{T} x_{1:T}$ is the average iterate,

$$f(\bar{x}_T) + \phi(\bar{x}_T) - f(x^*) - \phi(x^*) \leq \frac{1}{T} R_T^{(f+\phi)} .$$

In addition, if $f$ is non-convex but $\bar{x}_T$ is selected uniformly at random from $x_1, \ldots, x_T$, then the above bound holds in expectation. As such, in the rest of the paper we study the optimization risk through the lens of online regret.

**Stochastic first-order oracle.** Throughout the paper, we assume that at time $t$, the noisy gradient estimate $g_t$ is given by a randomized first-order oracle[3] $g_t : \mathbb{R}^d \times \Xi \to \mathbb{R}^d$, where $\Xi$ is some space of random variables, and there exists a sequence $(\xi_t)_{t=1}^{T}$ of independent elements from $\Xi$, with distribution $\mathbb{P}_\Xi$, such that $\int_\Xi g_t(x, \xi) d\mathbb{P}_\Xi(\xi) = \nabla f_t(x)$ for all $x \in \mathcal{X}$.

For example, in the finite-sum stochastic optimization case when $f = \sum_i^N f_i$, selecting one $f_i$ uniformly at random to estimate the gradient corresponds to $\mathbb{P}_\Xi$ being the uniform distribution on $\Xi = \{1, 2, \ldots, N\}$ and $g_t(x, \xi_t) = \nabla f_{\xi_t}(x)$, whereas selecting a mini-batch of $f_i$'s corresponds to $\Xi$ being the set of subsets (of a fixed or varying size) of $\{1, 2, \ldots, N\}$ and $g_t(x, \xi_t) = \frac{1}{|\xi_t|} \sum_{i \in \xi_t} \nabla f_i(x)$. This also covers variance-reduced gradient estimates as formed, e.g., by SAGA and SVRG, in which case $g_t$ is built using information from the previous rounds.[4]

# 3 Preliminaries: analysis in the serial setting

First, we recall the analysis of a generic *serial* dual-averaging algorithm, known as Adaptive Follow-the-Regularized-Leader (ADA-FTRL) [21, 25, 16], that generalizes regularized dual-averaging [37] and captures the dual-averaging variants of SGD, Ada-Grad, Proximal-SGD and EG as special case.

**Serial ADA-FTRL.** The serial ADA-FTRL algorithm uses a sequence of regularizer functions $r_0, r_1, r_2, \ldots$. At time $t = 1, 2, \ldots$, given the previous feedback $g_s \in \mathbb{R}^d, s \in [t-1]$, ADA-FTRL selects the next point $x_t$ such that

$$x_t \in \arg\min_{x \in \mathcal{X}} \langle z_{t-1}, x \rangle + t\phi(x) + r_{0:t-1}(x) , \tag{2}$$

where $z_{t-1} = g_{1:t-1}$ is the sum of the past feedback. We refer to $(z_t, t, r_{0:t})$ as the *state* of the algorithm at time $t$, noting that apart from tie-breaking in (2), this state determines $x_t$.

It is straightforward to verify that with $\phi = 0$, $\mathcal{X} = R^d$, and $r_{0:t-1} = \frac{\eta}{2} \| \cdot \|^2$ for some $\eta > 0$, we get the SGD update $x_t = -\frac{1}{\eta} g_{1:t-1}$. In addition, using $r_{0:t-1} = \frac{1}{2} \| \cdot \|_{\eta_t}^2$ where $\eta_t^{(i)}, i \in [d]$ are positive step-sizes (possibly adaptively tuned [22, 9]), ADA-FTRL reduces to $x_t = \mathbf{prox}(t\phi, -z_{t-1}, \eta_t)$, where $\mathbf{prox}$ is the generalized *proximal operator* oracle[5] over $\mathcal{X}$ that, given a function $\psi$ and vectors $z$ and $\eta$, returns[6]

$$\mathbf{prox}(\psi, z, \eta) := \arg\min_{x \in \mathcal{X}} \psi(x) + \frac{1}{2} \left\| x - \eta^{-1} \odot z \right\|_\eta^2 . \tag{3}$$

When $\eta$ is the same for all coordinates (in which case we simply treat it as a scalar), this reduces to $\mathbf{prox}(\psi, z, \eta) = \arg\min_{x \in \mathcal{X}} \psi(x) + \frac{\eta}{2}\|x - z/\eta\|^2$, which is the standard proximal operator; the generalized version (3) makes it possible to use coordinatewise step-sizes as in ADAGRAD [22, 9]. Finally, when $\phi = 0$ and $\mathcal{X}$ is the probability simplex, ADA-FTRL with the negentropy regularizer $r_{0:t-1}(x) = r_0(x) = \eta \sum_{i=1}^{d} x_i \log(x_i)$ for some $\eta > 0$, recovers the update $x_t^{(i)} = C_t \exp(-z_{t-1}^{(i)}/\eta)$ of the EG algorithm, where $C_t = 1/\sum_{j=1} \exp(-z_{t-1}^{(j)}/\eta)$ is the constant normalizing $x_t$ to lie in $\mathcal{X}$. Other choices of $r_t$ recover algorithms such as the $p$-norm update; we refer to Shalev-Shwartz [34], Hazan [14], McMahan [21], and Orabona et al. [25] for further examples.

**Analysis of ADA-FTRL** ADA-FTRL and its special cases have been extensively studied in the literature [5, 34, 14, 21, 25, 16]. In particular, it has been shown that under specific conditions on $r_t$ and $\phi$, which we discuss in detail in Appendix F, ADA-FTRL enjoys the following bound on the linearized regret [25, 16]:

**Theorem 1** (Regret of ADA-FTRL). *For any $x^* \in \mathcal{X}$ and any sequence of vectors $(g_t)_{t=1}^{T}$ in $\mathbb{R}^d$, using any sequence of regularizers $r_0, r_1, \ldots, r_T$ that are* admissible *w.r.t. a sequence of norms $\|\cdot\|_{(t)}$ (see Definition 2 in Appendix F), the iterates $(x_t)_{t=1}^{T}$ generated by ADA-FTRL satisfy*

$$\sum_{t=1}^{T} \left(\langle g_t, x_t - x^* \rangle + \phi(x_t) - \phi(x^*)\right) \leq r_{0:T}(x^*) - \sum_{t=0}^{T} r_t(x_{t+1}) + \sum_{t=1}^{T} \frac{1}{2}\|g_t\|_{(t,*)}^2 . \quad (4)$$

Importantly, this bound holds for *any* feedback sequence $g_t$ irrespective of the way it is generated, and serves as a solid basis to derive bounds under different assumptions on $f$, $\phi$, and $r_t$ [25, 16].

## 4 Relaxing the serial analysis: algorithms with perturbed state

In this section, we show that Theorem 1 can be used to analyze ADA-FTRL when its state undergoes specific perturbations. This relaxation of the generic serial analysis framework underlies our analysis of parallel asynchronous algorithms, since parallel algorithms like ASYNCADA and HEDGEHOG can be viewed as *serial* ADA-FTRL algorithms with perturbed states, as we show in Sections 5 and 6.

**Perturbed ADA-FTRL.** Next, we show that Theorem 1 also provides the basis to analyze ADA-FTRL with perturbed states. Specifically, suppose that instead of (2), the iterate $x_t$ is given by

$$x_t \in \arg\min_{x \in \mathcal{X}} \langle \hat{z}_{t-1}, x \rangle + \hat{t}_t \phi(x) + \hat{r}_{0:t-1}(x), \qquad t = 1, 2, \ldots, \quad (5)$$

where $\hat{z}_{t-1}$ denotes a *perturbed* version of the dual vector $z_{t-1}$, $\hat{t}_t$ denotes a perturbed version of ADA-FTRL's iteration counter $t$, and $\hat{r}_{0:t-1}$ denotes a perturbed version of the regularizer $r_{0:t-1}$. Then, we can analyze the regret of the Perturbed-ADA-FTRL update (5) by comparing $x_t$ to the "ideal" iterate $\tilde{x}_t$, given by

$$\tilde{x}_t := \arg\min_{x \in \mathcal{X}} \langle z_{t-1}, x \rangle + t\phi(x) + r_{0:t-1}(x), \qquad t = 1, 2, \ldots. \quad (6)$$

Since $(\tilde{x}_t)_{t=1}^{T}$ is given by a non-perturbed ADA-FTRL update, it enjoys the bound of Theorem 1. The crucial observation of Duchi et al. [10] (who studied the special case of (5) with $\phi = 0$, box-shaped $\mathcal{X}$, and $\hat{r}_t = r_t$) was that the regret of Perturbed-ADA-FTRL is related to the linearized regret of $\tilde{x}_t$. When $\phi$ may be non-zero, we capture this relation by the next lemma, proved in Appendix A:

**Lemma 1** (Perturbation penalty of ADA-FTRL). *Consider any sequences $(x_t)_{t=1}^{T}$ and $(\tilde{x}_t)_{t=1}^{T}$ in $\mathcal{X}$, and any sequence $(g_t)_{t=1}^{T}$ in $\mathbb{R}^d$. Then, the regret $R_T^{(f+\phi)}$ of the sequence $(x_t)_{t=1}^{T}$ satisfies*

$$R_T^{(f+\phi)} = \sum_{t=1}^{T} \left(\langle g_t, \tilde{x}_t - x^* \rangle + \phi(\tilde{x}_t) - \phi(x^*)\right) + \tilde{\epsilon}_{1:T} + \delta_{1:T} - B_{1:T} , \quad (7)$$

*where $\tilde{\epsilon}_t = \langle g_t, x_t - \tilde{x}_t \rangle + \phi(x_t) - \phi(\tilde{x}_t)$, $\delta_t = \langle \nabla f_t(x_t) - g_t, x_t - x^* \rangle$ and $B_t = \mathcal{B}_{f_t}(x^*, x_t)$.*

Since $g_t$ is an unbiased estimate of $\nabla f_t(x_t)$ (conditionally given $x_t$), $\delta_{1:T}$ is zero in expectation 0, and for $\tilde{x}_t$ given by (6), the first summation is bounded by Theorem 1. Also note that when the $f_t$ are (star-)convex, $-B_{1:T} \leq 0$. Thus, to bound the regret of Perturbed-ADA-FTRL, it only

remains to control the "perturbation penalty" terms $\tilde{\epsilon}_t$ capturing the difference in the composite linear loss $\langle g_t, \cdot \rangle + \phi$ between $x_t$ and $\tilde{x}_t$. In Appendix A, we use the stability of ADA-FTRL algorithms (Lemma 3) to control $\tilde{\epsilon}_{1:T}$, under a specific perturbation structure (coming from delayed updates to $\hat{z}_t$) that captures the evolution of the state of asynchronous dual-averaging algorithms like ASYNCADA and HEDGEHOG. Unlike Duchi et al. [10], our derivation applies to *any* convex constraint set $\mathcal{X}$ and, crucially, to ADA-FTRL updates incorporating non-zero $\phi$ and a perturbed counter $\hat{t}_t$. The following (informal) theorem, whose formal version is given in Appendix A, captures the result.

**Theorem 4 (informal).** Under appropriate independence, regularity, and structural assumptions on the regularizers and the perturbations, the Perturbed-ADA-FTRL update (5) satisfies

$$\mathbb{E}\left\{ R_T^{(f+\phi)} \right\} \leq \mathbb{E}\left\{ r_{0:T}(x^*) + \sum_{t=1}^{T} \left( \frac{1 + p_* \nu_t + \sum_{s:t \in O_s} \frac{\tau_s}{\nu_s}}{2} \|g_t\|_{(t,*)}^2 + \frac{\Delta_t}{\nu_t} \right) - B_{1:T} \right\},$$

where $p_*, \nu_t, \tau_t$ and $\Delta_t$ measure, respectively, the sparsity of the gradient estimates $g_t$, the difference $\hat{t}_t - t$, and the amount of perturbations in $\hat{z}_{t-1}$, and $\hat{r}_{0:t-1}$, while $O_s$ is the set of time steps whose attributed perturbations affect iteration $s$ (i.e., their updates are delayed beyond $s$).

As we show next, we can control the effect of $p_*, \tau_t$ and $\Delta_t$ in the bound by appropriately tuning $\hat{t}_t$, resulting in linear speed-ups for ASYNCADA and HEDGEHOG.

## 5 ASYNCADA: Asynchronous Composite Adaptive Dual Averaging

In this section, we introduce and analyze ASYNCADA for asynchronous noisy online optimization. ASYNCADA consists of $\tau$ processes running in parallel (e.g., threads on the same physical machine or computing nodes distributed over a network accessing a shared data store). The processes can access a shared memory, consisting of a *dual* vector $z \in \mathbb{R}^d$ to store the sum of observed gradient estimates $g_t$, a *step-size* vector $\eta \in \mathbb{R}^d$, and an integer $t$, referred to as the *clock*, to track the number of iterations completed at each point in time. The processes run copies of Algorithm 1 concurrently.

---

**Algorithm 1:** ASYNCADA: Asynchronous Composite Adaptive Dual Averaging

---
1 **repeat**
2    $\hat{\eta} \leftarrow$ a full (lock-free) read of the shared step-sizes $\eta$
3    $\hat{z} \leftarrow$ a full (lock-free) read of the shared dual vector $z$
4    $t \leftarrow t + 1$                          `// atomic read-increment`
5    $\hat{t} \leftarrow t + \gamma$                 `// denote` $\hat{z}_{t-1} = \hat{z}$, $\hat{\eta}_t = \hat{\eta}, \hat{t}_t = \hat{t}$
6    Receive $\xi_t$
7    Compute the next iterate: $x_t \leftarrow \mathbf{prox}(\hat{t}_t \phi, -\hat{z}_{t-1}, \hat{\eta}_t)$     `// prox defined in (3)`
8    Obtain the noisy gradient estimate: $g_t \leftarrow g_t(x_t, \xi_t)$
9    **for** $j$ such that $g_t^{(j)} \neq 0$ **do** $z^{(j)} \leftarrow z^{(j)} + g_t^{(j)}$       `// atomic update`
10    Update the shared step-size vector $\eta$
11 **until terminated**

---

**Inconsistent reads.** The processes access the shared memory without necessarily acquiring a lock: as in previous Hogwild!-style algorithms [30, 20, 18, 17, 27], we only assume that operations on single coordinates of $z$ and $\eta$, as well as on $t'$, are atomic. This in particular means that the values of $\hat{z}$ or $\hat{\eta}$ read by a process may not correspond to an actual state of $z$ or $\eta$ at any given point in time, as different processes can modify the coordinates in parallel while the read is taking place. A process $\pi$ is in write-conflict with another process $\pi'$ (equivalently, $\pi'$ is in read-conflict with $\pi$) if $\pi'$ reads parts of the memory which should have been updated by $\pi$ before. To limit the effects of asynchrony, we assume that a process can be in write- and read conflicts with at most $\tau_c - 1$ processes, respectively.

**The role of $\gamma$.** ASYNCADA uses an over-estimate $\hat{t}_t$ of the current global clock $t$ by an additional $\gamma$. This over-estimation enables us to better handle the effect of asynchrony when composite objectives are involved, in particular ensuring the appropriate tuning of $\nu_t$ in Theorem 4; see Appendix C. ASYNCADA can nevertheless be run without $\gamma$ (i.e., with $\gamma = 0$).[7]

**Exact vs estimated clock.** ASYNCADA as given in Algorithm 1 maintains the exact global clock $t$. However, this option may not be desirable (or available) in certain asynchronous computing scenarios. For example, if the processes are distributed over a network, then maintaining an exact global clock amounts to changing the pattern of asynchrony and delaying the computations by repeated calls over a network. To mitigate this requirement, in Appendix B we provide ASYNCADA($\rho$), a version of ASYNCADA in which the processes update the global clock only every $\rho$ iterations. ASYNCADA as presented in Algorithm 1 is equivalent to ASYNCADA($\rho$) with $\rho = 1$, and both algorithms enjoy the same rate of convergence and linear speed-up. Obviously, when $\phi \equiv 0$ and $t$ is not used for setting the step-sizes $\eta$ either, there is no need to maintain $t$ physically, and Line 4 can be omitted.

**Updating the step-sizes $\eta$:** In Line 10 of Algorithm 1, the step-size $\eta$ has to be updated based on the information received. The exact way this is done depends on the specific step-size schedule. In particular, we consider two situations: First, when the step-size is either constant or a simple function of $t$ (or $\hat{t}_t$ in case of ASYNCADA($\rho$)), and second, when diagonal ADA-GRAD step-sizes are used. In the first case, the vector $\eta$ need not be kept in the shared memory explicitly, and Lines 2 and 10 can be omitted. In the second case, following [10], we store the sum of squared gradients in the shared $\eta$, i.e., Line 10 is implemented as follows:

---

10* **for** $j$ such that $g_t^{(j)} \neq 0$ **do** $\left(\eta^{(j)}\right)^2 \leftarrow \left(\eta^{(j)}\right)^2 + \alpha^2 \left(g_t^{(j)}\right)^2$     // `atomic update`

---

for a fixed hyper-parameter $\alpha > 0$. In this case, we are storing the square of $\eta$ in the shared memory, so a square root operation needs to be applied after reading the shared memory in Line 2 to retrieve $\eta$.

**Forming the output $\bar{x}_T$ for stochastic optimization:** For stochastic optimization, the algorithm needs to output the average (or randomized) iterate $\bar{x}_T$ at the end. However, this needs no further coordination between the processes. To form the average iterate, it suffices for each process to keep a local running sum of the iterates it produces and the number of updates it makes. At the end, $\bar{x}_T$ is built from these sums and the total number of updates. Alternatively, we can return a random iterate as $\bar{x}_T$ by terminating the algorithm, with probability $1/T$, after calculating $x$ in Line 7.

## 5.1 Analysis of ASYNCADA

The analysis of ASYNCADA is based on treating it as a special case of Perturbed-ADA-FTRL. In order to be able to use Theorem 4, we start with the following independence assumption on $\xi_t$:

**Assumption 1** (Independence of $\xi_t$). For all $t = 1, 2, \ldots, T$, the $t$-th sample $\xi_t$ is independent of the history $\hat{\mathcal{H}}_t := \left\{ (\xi_s, \hat{z}_s, \hat{\eta}_{s+1})_{s=1}^{t-1} \right\}$.

This, in turn, implies that $\xi_t$ is independent of $x_t$ as well as $x_s$ and $\xi_s$ for all $s < t$.

For general (non-box-shaped) $\mathcal{X}$, Assumption 1 is plausible, as ASYNCADA *needs* to read $z$ (and $\eta$) completely and independently of $\xi_t$. If $\mathcal{X}$ is box-shaped and $\phi$ is coordinate-separable, however, the values of $x_t^{(j)}$ for different coordinates $j$ can be calculated independently. In this, case, the algorithm may first sample $\xi_t$, and then only read the relevant coordinates $j$ from $z$ (and $\eta$) for which $g_t$ may be non-zero, as calculating other values of $x_t^{(j)}$ is unnecessary for calculating $g_t$. As mentioned by Mania et al. [20], this violates Assumption 1. This is because multiple other processes are updating $z$ and $\eta$, and the updates that are included the value read for $\hat{z}_{t-1}$ (and $\hat{\eta}_t$) would then depend on $\xi_t$. Previous papers either assume that this independence holds in their analysis, e.g., by enforcing a full read of $z$ and $\eta$, [20, 18, 17, 27], or rely on the smoothness of the objective to bound the effect of the possible change in the read values [20, Appendix A]. It seems possible to adapt the argument of Mania et al. [20, Appendix A] to ASYNCADA for box-shaped $\mathcal{X}$, by comparing $x_t$ to the iterate that would have been created based on the content of the shared memory right before the start of the execution of the $t$-th iteration. This makes the analysis more complicated, and is not necessary when $\mathcal{X}$ is not box-shaped; hence, we do not further pursue this construction in this paper.

**Sparsity of the gradient estimates.** For $t \in [T]$ and $j \in [d]$, let $p_{t,j}$ to denote the probability that the $j$-th coordinate of $g_t$ is non-zero given the history $\hat{\mathcal{H}}_t$, that is, $p_{t,j} = \mathbb{P}\{g_t^{(j)} \neq 0 | \hat{\mathcal{H}}_t\}$. Let $p_*$ denote an upper-bound on $\max_{t \in [T], j \in [d]} p_{t,j}$. We use $p_*$ as a measure of the sparsity of the problem.[8]

**Non-adaptive and time-decaying step-sizes.** We first study the case when $\eta_t$ is either a constant, or varies only as a function of the estimated iteration count $\hat{t}_t$. Recall that each concurrent iteration of the algorithms can be in read- and write-conflict with at most $\tau_c - 1$ other iterations, respectively, and that the algorithm uses $\tau$ parallel processes. Define $\tau_* = \max\{\tau_c, \tau\}$. The next theorem gives bounds on the regret of ASYNCADA under various scenarios. It is proved in Appendix C, where a similar result is also given for ASYNCADA($\rho$) (Theorem 5).

**Theorem 2.** *Suppose that either all $f_t, t \in [T]$ are convex, or $\phi \equiv 0$ and $f_t \equiv f$ for some star-convex function $f$. Consider* ASYNCADA *running under Assumption 1 for $T > \tau_*^2$ updates, using $\gamma = 2\tau_*^2$. Let $\eta_0 > 0$. Then:*

*(i) If $\mathbb{E}\{\|g_t\|_2^2\} \leq G_*^2$ for all $t \in [T]$, then using a fixed $\eta_t = \eta_0\sqrt{T}$ or a time-varying $\eta_t = \eta_0\sqrt{\hat{t}_t}$,*

$$\frac{1}{T}\mathbb{E}\left\{R_T^{(f+\phi)}\right\} \leq \frac{1}{\sqrt{T}}\left(\eta_0\|x^*\|_2^2 + \frac{2(1+p_*\tau_*^2)}{\eta_0}G_*^2\right) . \qquad (8)$$

*(ii) If $f_t = f = \mathbb{E}_{\xi\sim\mathbb{P}_\Xi}\{F(x,\xi)\}$, $\sigma_*^2 := \mathbb{E}\{\|\nabla F(x^*,\cdot)\|_2^2\}$, and for all $\xi \in \Xi$, $F(\cdot,\xi)$ is convex and 1-smooth w.r.t. the norm $\|\cdot\|_l$ for some $l \in \mathbb{R}^d$ with positive entries, then given a constant $c_0 > 8(1+p_*\tau_*^2)$ and using a fixed $\eta_{t,i} = c_0l_i + \eta_0\sqrt{T}$ or a time-varying $\eta_{t,i} = c_0l_i + \eta_0\sqrt{\hat{t}_t}$,*

$$\frac{1}{T}\mathbb{E}\left\{R_T^{(f+\phi)}\right\} \leq \frac{c_0\|x^*\|_l^2}{T} + \frac{2}{\sqrt{T}}\left(\eta_0\|x^*\|_2^2 + \frac{4(1+p_*\tau_*^2)}{\eta_0}\sigma_*^2\right) . \qquad (9)$$

*(iii) If $\phi$ is $\mu$-strongly-convex and $\mathbb{E}\{\|g_t\|_2^2\} \leq G_*^2$ for all $t \in [T]$, then using $\eta_t \equiv 0$ or, equivalently, $\mathbf{prox}(\hat{t}_t\phi, -z, 0) := \arg\min_{x\in\mathcal{X}} \hat{t}_t\phi(x) + \langle z, x\rangle = \nabla\phi^*(-z/\hat{t}_t)$,*

$$\frac{1}{T}\mathbb{E}\left\{R_T^{(f+\phi)}\right\} \leq \frac{(1+p_*\tau_*^2)G_*^2(1+\log(T))}{\mu T} . \qquad (10)$$

**Remark 1.** If $c = p_*\tau_*^2$ is constant, the bounds match the corresponding serial bounds [16] up to constant factors, implying a linear speed-up. This also extends the analysis of ASYNC-DA [10] to non-box-shaped $\mathcal{X}$, non-zero $\phi$, time-varying step sizes, and smooth and strongly-convex objectives.[9]

**Remark 2.** Note that (10) holds for all time steps, and converges to zero as $T$ grows, without the knowledge of $T$ or epoch-based updates. In case of ASYNCADA($\rho$), the algorithm does not maintain an exact clock either. To our knowledge, this makes ASYNCADA($\rho$) the first Hogwild!-style algorithm with an any-time guarantee without maintaining a global clock.

**Remark 3.** Since strongly convex functions have unbounded gradients on unbounded domains, it is not possible to impose a uniform bound on the gradient of $f + \phi$ in part (iii) for unconstrained optimization (i.e., when $\mathcal{X} = \mathbb{R}^d$). However, we only require the gradients of $f$, the non-strongly-convex part of the objective, to be bounded, which is a feasible assumption. Similarly, Nguyen et al. [24] analyzed strongly-convex optimization with unconstrained Hogwild! while avoiding the aforementioned uniform boundedness assumption,s using a global clock. ASYNCADA($\rho$) achieves the same result, but applies to arbitrary convex $\mathcal{X}$ and $\phi$, without requiring a global clock.

**Adaptive step-sizes.** Due to space constraints, we relegate the analysis of ASYNCADA($\rho$) with AdaGrad step-sizes given by Line 10* to Appendix D.

# 6 HEDGEHOG: Hogwild-Style Hedge

Next, we present HEDGEHOG, which is, to our knowledge the first asynchronous version of the EG algorithm. The parallelization scheme is very similar to ASYNCADA, the difference being that EG uses multiplicative updates rather than additive SGD-style updates. We focus only on the case of $\phi \equiv 0$. Each processe runs Lines 3–10 of Algorithm 2 concurrently with the other processes, sharing the dual vector $z$.

---

sparsity of the problem through a "conflict graph" [30, 20, 17, 27], which is a bi-partite graph with $f_i, i \in [m]$ on the left and coordinates $j \in [d]$ on the right, and an edge between $f_i$ and coordinate $j$ if $\nabla f_i(x)^{(j)}$ can be non-zero for some $x \in \mathcal{X}$. In this graph, let $\delta_j$ denote the degree of the node corresponding to coordinate $j$ and $\Delta_r$ be the largest $\delta_j, j \in [d]$. Then, it is straightforward to see that $p_{t,j} \leq \delta_j/m$. Thus, $p_* = \Delta_r/m$ is a valid upper-bound, and gives the sparsity measure used, e.g., by Leblond et al. [17] and Pedregosa et al. [27].

[9]Note that under the conditions considered in [10], which include that $\mathcal{X}$ is box-shaped and $\phi = 0$, ASYNC-DA requires a less restrictive sparsity regime of $p_*\tau_* \leq c$ for linear speed-up.

**Algorithm 2:** HEDGEHOG!: Asynchronous Stochastic Exponentiated Gradient.

**Input:** Step size $\eta$
1  **Initialization**
2  $\quad$ Let $z \leftarrow 0$ be the shared sum of observed gradient estimates
3  **repeat in parallel by each process**
4  $\quad$ $\hat{z} \leftarrow$ a full lock-free read of the shared dual vector $z$ $\qquad$ // $t \leftarrow t+1$, denote $\hat{z}_{t-1} = \hat{z}$
5  $\quad$ Receive $\xi_t$
6  $\quad$ Compute the next iterate: $w_t^{(i)} \leftarrow \exp\left(-\hat{z}_{t-1}^{(i)}/\eta\right), \qquad i = 1, 2, \ldots, d$
7  $\quad$ Normalize: $x_t \leftarrow w_t/\|w_t\|_1$
8  $\quad$ Obtain the noisy gradient estimate: $g_t \leftarrow g_t(x_t, \xi_t)$
9  $\quad$ **for** $j$ such that $g_t^{(j)} \neq 0$ **do** $z^{(j)} \leftarrow z^{(j)} + g_t^{(j)}$ $\qquad$ // atomic update
10 **until terminated**

As in ASYNCADA($\rho$), we index the iterations by the time they finish the reading of $z$ in Line 4 of HEDGEHOG ("after-read" labeling [18]). Similarly, we use $\hat{\mathcal{H}}_t = \left\{(\xi_s, \hat{z}_s)_{s=1}^{t-1}\right\}$ to denote the history of HEDGEHOG at time $t$, and use $\hat{\mathcal{H}}_t$ to define the sparsity measure $p_*$ as in Section 5.1. Then, we have the following regret bound for HEDGEHOG.

**Theorem 3.** *Let $\mathcal{X}$ be the probability simplex $\mathcal{X} = \{x|x^{(j)} > 0, \|x\|_1 = 1\}$, and suppose that either $f_t$ are all convex, or $f_t \equiv f$ for a star-convex $f$. Assume that for all $t \in [T]$, the sampling of $\xi_t$ in Line 5 of HEDGEHOG is independent of the history $\hat{\mathcal{H}}_t$. Then, after $T$ updates, HEDGEHOG satisfies*

$$\mathbb{E}\left\{R_T^{(f)}\right\} \leq \eta \log(d) + \sum_{t=1}^{T} \mathbb{E}\left\{\frac{1 + \sqrt{p_*}\tau_*}{2\eta}\|g_t\|_\infty^2\right\}.$$

**Remark 4.** As in the case of ASYNCADA, as long as $\sqrt{p_*}\tau_*$ is a constant, the rate above matches the worst-case rate of serial EG up to constant factors, implying a linear speed-up. In particular, given an upper-bound $G_*$ on $\mathbb{E}\{\|g_t\|_\infty\}$ and setting $\eta = G_*/\sqrt{T\log(d)}$, we recover the well-known $\mathcal{O}(G_*\sqrt{T\log(d)})$ rate for EG [14], but in the paralell asynchronous setting.

## 7  Conclusion, limitations, and future work

We presented and analyzed ASYNCADA, a parallel asynchronous online optimization algorithm with composite, adaptive updates, and global convergence rates under generic convex constraints and convex composite objectives which can be smooth, non-smooth, or non-strongly-convex. We also showed a similar global convergence for the so-called "star-convex" class of non-convex functions. Under all of the aforementioned settings, we showed that ASYNCADA enjoys linear speed-ups when the data is sparse. We also derived and analyzed HEDGEHOG, to our knowledge the first Hogwild-style asynchronous variant of the Exponentiated Gradient algorithm working on the probability simplex, and showed that HEDGEHOG enjoyed similar linear speed-ups.

To derive and analyze ASYNCADA and HEDGEHOG, we showed that the idea of perturbed iterates, used previously in the analysis of asynchronous SGD algorithms, naturally extends to generic dual-averaging algorithms, in the form of a perturbation in the "state" of the algorithm. Then, building on the work of Duchi et al. [10], we studied a unified framework for analyzing generic adaptive dual-averaging algorithms for composite-objective noisy online optimization (including ASYNCADA and HEDGEHOG as special cases). Possible directions for future research include applying the analysis to other problem settings, such as multi-armed bandits. In addition, it remains an open problem whether such an analysis is obtainable for constrained adaptive Mirror Descent without further restrictions on the regularizers (e.g., smoothness of the regularizer seems to help). Finally, the derivation of such data-dependent bounds for the final (rather than the average) iterate in stochastic optimization, without the usual strong-convexity and smoothness assumptions, remains an interesting open problem.

## Footnotes

[2] In this paper, we do not further consider BCD-based methods, for two main reasons: a) in general, a BCD update may unnecessarily slow down the convergence of the algorithm by focusing only on a single coordinate of the gradient information, especially in the sparse-data problems we consider in this paper (see, e.g., Pedregosa et al. [27, Appendix F]); and b) BCD algorithms typically apply only to box-shaped constraints, which is what our algorithms are designed to be able to avoid. We would like to note, however, that our stochastic gradient oracle set-up (Section 2) does allow for building an unbiased gradient estimate using only one randomly-selected (block-)coordinate, as done in BCD methods. Nevertheless, the literature on parallel asynchronous BCD algorithms is vast, including especially algorithms for proximal, non-strongly-convex, and non-convex optimization; see, e.g., [29, 11, 4, 2, 3, 19, 31, 33, 32, 35, 36, 12, 6, 28] and the references therein.

[3]With a slight abuse of notation, $g_t(x, \xi)$ (with arguments $x, \xi$) is from now on used to denote the oracle at time $t$ evaluated at $x, \xi$, where as $g_t$ (without arguments) denotes the observed noisy gradient $g_t(x_t, \xi_t)$.

[4]Note that in this case $\xi_t$ remains an independent sequence, even though $g_t$ changes with the history.

[5] Serial proximal DA [37] and ADA-FTRL call $\mathbf{prox}$ with $\psi \leftarrow t\phi$, whereas the conventional Proximal-SGD algorithm (based on Mirror-Descent) invokes the proximal operator with $\psi \leftarrow \phi$ irrespective of the iteration; see the paper of Xiao [37, Sections 5 and 6] for a detailed discussion of this phenomenon.

[6]Here $\eta^{-1}$ denotes the elementwise inverse of $\eta$ and $\odot$ denotes elementwise multiplication.

[7] In Theorems 2, 5 and 6, we set $\gamma$ based on $\tau_* := \max\{\tau_c, \tau\}$. The analysis is still possible, and straightforward, with $\gamma = 0$, but results in a worst constant factor in the rate, as well as an extra additive term of order $\mathcal{O}(\tau_*^2 \Phi)$ where $\Phi = \sup_{x,y \in \mathcal{X}} \{\phi(x) - \phi(y)\}$ is the diameter of $\mathcal{X}$ w.r.t. $\phi$. This term does not diminish with $p_*$ and may be unnecessarily large, affecting convergence in early stages of the optimization process.

[8] In stochastic optimization with a finite-sum objective $f = \sum_{i=1}^m f_i$, where $g_t = \nabla f_{\xi_t}(x_t)$ and $\xi_t \in [m]$ is an index at time $t$ sampled uniformly at random and independently of the history, one could measure the

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
