[Supplementary Material]

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

[10]Proposition 1 of Duchi et al. [10] establishes, for $\phi = 0$, a lower-bound of $\frac{1}{8} R \sum_{j=1}^d G_j \min\left\{p_j, \sqrt{p_j/T}\right\}$. By Theorem 6, ASYNCADA($\rho$) with ADAGRAD step-sizes matches this rate (when $\phi = 0$), since in addition to (24), by the sparsity of the gradients and star-convexity of $f$, we have for any $x \in \mathcal{X}$ and $\xi$ sampled from $\mathbb{P}_\Xi$ that

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

# A Proofs for the generic framework

*Proof of Lemma 1.* The proof follows in the same way as in the serial setting [16]. For $t \in [T]$,

$$
\begin{aligned}
f_t(x_t) - f_t(x^*) &= \langle \nabla f_t(x_t), x_t - x^* \rangle - \mathcal{B}_{f_t}(x^*, x_t) \\
&= \langle g_t, x_t - x^* \rangle + \langle \nabla f_t(x_t) - g_t, x_t - x^* \rangle - \mathcal{B}_{f_t}(x^*, x_t) \\
&= \langle g_t, \tilde{x}_t - x^* \rangle + \langle g_t, x_t - \tilde{x}_t \rangle + \delta_t - B_t \\
&= \langle g_t, \tilde{x}_t - x^* \rangle + \phi(\tilde{x}_t) - \phi(x_t) + \tilde{\epsilon}_t + \delta_t - B_t
\end{aligned}
$$

Adding $\phi(x_t) - \phi(x^*)$ to both sides and summing over $t$ completes the proof. $\square$

**Perturbation structure.** We assume that the difference of $\hat{z}_{t-1}$ and $z_{t-1}$ is that zero or more coordinates $g_s^{(j)}$ from the past feedback vectors $g_s, s \in [t-1]$, can be missing from (i.e., not added in) the perturbed dual vector $\hat{z}_{t-1}$. Formally, for all $t \in [T]$ and $j \in [d]$,

$$
\hat{z}_{t-1}^{(j)} = g_{1:t-1}^{(j)} - \sum_{s \in O_{t,j}} g_s^{(j)} , \tag{11}
$$

where $O_{t,j}$ is the subset of the past indices $[t-1]$ corresponding to the missing updates at the $j$-th coordinate. Written in a more compact form,

$$
\hat{z}_{t-1} = g_{1:t-1} - \sum_{s \in O_t} I_{t,s} g_s , \tag{12}
$$

where $O_t = \cup_j O_{t,j}$ is the set of all time steps with missing information at time $t$ (that is, the set of iterations with which iteration $t$ is in read-conflict), and $I_{t,s}, s \in [t-1]$, are diagonal $d \times d$ matrices with $I_{t,s}^{(j,j)} = 1$ if $g_s^{(j)}$ is missing from $\hat{z}_{t-1}$ and 0 otherwise. We define $\tau_{t,j} = |O_{t,j}|$ and $\tau_t = |O_t|$ to denote, respectively, the total number of missing updates to the $j$-th coordinate of $\hat{z}_{t-1}$, and to the whole vector $\hat{z}_{t-1}$. Similarly, we assume that the time-counter $\hat{t}_t$ may not be equal to $t$, and the cumulative regularizers $r_{0:t}$ and $\hat{r}_{0:t}$, can be different, with the latter using only some of the past updates made to $r_{0:t}$. However, the exact perturbation in $\hat{t}_t$ and $\hat{r}_{0:t}$ depends on the specifics of the algorithm. Our analysis isolates these perturbations in individual terms, which we can subsequently study on a case-by-case basis. We make the following assumption on $\hat{t}_t$ and the sequence of actual regularizers $(\hat{r}_t)_{t=0}^T$ and ideal regularizers $(r_t)_{t=0}^T$.

**Assumption 2.** The regularizers $r_t, \hat{r}_t, t = 0, 1, \ldots, T$, are admissible ADA-FTRL regularizers (Definition 2) with the same sequence of norms $\|\cdot\|_{(t)}$, and the sequence of norms is non-decreasing: $\|\cdot\|_{(t)} \geq \|\cdot\|_{(t-1)}$ for all $t = 1, 2, \ldots, T$. Finally, $r_t \geq 0, t = 0, 1, 2, \ldots, T$, and $\hat{t}_t > t, t = 1, 2, \ldots, T$.

Intuitively, Assumption 2 states that the regularizers $\hat{r}_t$ are not fundamentally different from the regularizers $r_t$ as far as the basic properties of ADA-FTRL are concerned. In particular, the assumption is satisfied if $(r_t)_{t=0}^T$ is admissible with a non-decreasing sequence of norms and the perturbation increases the curvature, that is, $\hat{r}_{0:t-1} - r_{0:t-1}$ is convex. Finally, the assumption $\hat{t}_t > t$ helps us in providing bounds for composite-objective learning, as will become clear later.

**Independence assumption.** Similarly to the standard serial setting, we will assume that the outcome $\xi_t$ at time $t$ is independent of the history that determines $x_t$. In the case of perturbed ADA-FTRL, we define the history to depend on the actual states the *perturbed* ADA-FTRL algorithm has gone through:

**Definition 1** (History of the perturbed game). For $t = 1, 2, \ldots, T$, the *history of the perturbed game* up to time $t$ is defined as

$$
\hat{\mathcal{H}}_t = \left\{ \left( \xi_s, \hat{z}_s, \hat{t}_s, \hat{r}_{0:s} \right)_{s=1}^{t-1} \right\},
$$

where $\hat{z}_s, \hat{r}_{0:s}, \hat{t}_s$ are the dual vector, regularizer and time-counter used by the $(s+1)$-th perturbed ADA-FTRL update.

We assume that the stochastic outcomes are independent of the history:

**Assumption 3** (Independence of $\xi_t$). *For all $t = 1, 2, \ldots, T$, the $t$-th sample $\xi_t$ is independent of the history $\hat{\mathcal{H}}_t$.*

This in turn means that $\xi_t$ is independent of $x_t$ as well as $x_s$ and $\xi_s$ for all $s < t$.

We call a norm $\|\cdot\|$ a *weighted $q$-norm* if there exists $q > 0$ and $a_j, j \in [d]$ such that for all $x \in \mathbb{R}^d$,

$$\|x\| = \left( \sum_{j=1}^{d} a_j \left| x^{(j)} \right|^q \right)^{1/q} . \tag{13}$$

The next theorem describes a generic data-dependent bound on the regret of perturbed ADA-FTRL.

**Theorem 4.** *Suppose that Perturbed-ADA-FTRL is run under Assumption 3, and Assumption 2 holds such that for each $t \in [T]$, $\|\cdot\|_{(t)}$ is a weighted $q$-norm with $q = 1$ or $q = 2$. For all $t \in [T]$, define $\Delta_t = r_{0:t-1}(x_t) - r_{0:t-1}(\tilde{x}_t) + \hat{r}_{0:t-1}(\tilde{x}_t) - \hat{r}_{0:t-1}(x_t)$, and $\nu_t = \hat{t}_t - t$ with the $\hat{t}_t$ used in the Perturbed-ADA-FTRL update (5). Then, the regret of Perturbed-ADA-FTRL satisfies*

$$\mathbb{E}\left\{ R_T^{(f+\phi)} \right\} \leq \mathbb{E}\left\{ r_{0:T}(x^*) + \sum_{t=1}^{T} \left( \frac{1 + p_* \nu_t + \sum_{s:t \in O_s} \frac{\tau_s}{\nu_s}}{2} \|g_t\|_{(t,*)}^2 + \frac{\Delta_t}{\nu_t} \right) - B_{1:T} \right\},$$

*where $p_*$ is a global upper-bound on $\mathbb{P}\left\{ g_t^{(j)} \neq 0 | \hat{\mathcal{H}}_t \right\}$.*

## A.1 Proof of Theorem 4

First, we upper-bound $\tilde{\epsilon}_t$ in terms of the difference between $\tilde{x}_t$ and $x_t$.

**Lemma 2.** *Consider Perturbed-ADA-FTRL under the conditions of Theorem 4. Let $\beta_t \in \mathbb{R}^d$ be given by $\beta_t^{(j)} = \mathbb{I}\left\{ g_t^{(j)} \neq 0 \right\}$, and use $\odot$ to denote elementwise vector multiplication. Then,*

- *For any positive real number $c_t$ and any norm $\|\cdot\|$, we have*

$$\tilde{\epsilon}_t + \phi(\tilde{x}_t) - \phi(x_t) \leq \frac{c_t}{2} \|g_t\|_*^2 + \frac{1}{2c_t} \|\beta_t \odot (x_t - \tilde{x}_t)\|^2 ,$$

- *In the stochastic setting under Assumption 3, for any $c_t > 0$ and any norm $\|\cdot\|$,*

$$\mathbb{E}\{\tilde{\epsilon}_t + \phi(\tilde{x}_t) - \phi(x_t)\} \leq \mathbb{E}\left\{ \frac{c_t}{2} \|\nabla f_t(x_t)\|_*^2 + \frac{1}{2c_t} \|x_t - \tilde{x}_t\|^2 \right\} .$$

- *Under Assumption 3, for any $q \geq 1$, any weighted $q$-norm $\|\cdot\|$ determined by the history $\hat{\mathcal{H}}_t$, and any positive scalar $c_t \in \sigma(\hat{\mathcal{H}}_t)$,*

$$\mathbb{E}\{\tilde{\epsilon}_t + \phi(\tilde{x}_t) - \phi(x_t)\} \leq \mathbb{E}\left\{ \frac{c_t}{2} \|g_t\|_*^2 \right\} + p_*^{(1/q)} \mathbb{E}\left\{ \frac{1}{2c_t} \|(x_t - \tilde{x}_t)\|^2 \right\} ,$$

*where $p_*$ is a global upper-bound on $\mathbb{P}\left\{ g_t^{(j)} \neq 0 | \hat{\mathcal{H}}_t \right\}$. In case of $q = 2$, the bound still holds if $p_*^{1/2}$ is replaced with $p_*$.*

*Proof of Lemma 2.* To get the first inequality, note that $g_t = \beta_t \odot g_t$ by definition. The bound then follows by the Fenchel-Young inequality.

To get the second bound, note that $x_t, \tilde{x}_t \in \sigma(\hat{\mathcal{H}}_t)$ by construction, so by Assumption 3,

$$\mathbb{E}\{\langle g_t - \nabla f_t(x_t), x_t - \tilde{x}_t \rangle\} = \mathbb{E}\left\{ \langle \mathbb{E}\left\{ g_t - \nabla f_t(x_t) | \hat{\mathcal{H}}_t \right\}, x_t - \tilde{x}_t \rangle \right\} = 0 .$$

Thus, $\mathbb{E}\{\tilde{\epsilon}_t + \phi(\tilde{x}_t) - \phi(x_t)\} = \mathbb{E}\{\langle \nabla f_t(x_t), x_t - \tilde{x}_t \rangle\}$, and the result follows by the Fenchel-Young inequality.

To get the third bound, we first start with the simpler case of $q = 2$, using $a \in \sigma(\hat{\mathcal{H}}_t)$ to denote the associated weighting vector, then apply the first inequality and take expectation of the terms $\|\beta_t \odot (x_t - \tilde{x}_t)\|^2$. Note that by construction, $x_t, \tilde{x}_t \in \sigma(\hat{\mathcal{H}}_t)$. Furthermore, by assumption, $c_t, a \in \sigma(\hat{\mathcal{H}}_t)$. Hence,

$$
\begin{aligned}
\mathbb{E}\left\{\frac{1}{2c_t}\|\beta_t \odot (x_t - \tilde{x}_t)\|^2\right\} &= \mathbb{E}\left\{\mathbb{E}\left\{\frac{1}{2c_t}\|\beta_t \odot (x_t - \tilde{x}_t)\|^2\big|\hat{\mathcal{H}}_t\right\}\right\} \\
&= \mathbb{E}\left\{\sum_{j=1}^{d}\mathbb{E}\left\{\frac{1}{2c_t}a^{(j)}\beta_t^{(j)}(x_t^{(j)}-\tilde{x}_t^{(j)})^2|\hat{\mathcal{H}}_t\right\}\right\} \\
&= \mathbb{E}\left\{\sum_{j=1}^{d}\mathbb{E}\left\{\mathbb{I}\{g_t^{(j)}\neq 0\}|\hat{\mathcal{H}}_t\right\}\frac{1}{2c_t}a^{(j)}(x_t^{(j)}-\tilde{x}_t^{(j)})^2\right\} \\
&= \mathbb{E}\left\{\sum_{j=1}^{d}p_{t,j}\frac{1}{2c_t}a^{(j)}(x_t^{(j)}-\tilde{x}_t^{(j)})^2\right\} \\
&\leq \left(\max_{j\in[d]}p_{t,j}\right)\mathbb{E}\left\{\frac{1}{2c_t}\sum_{j=1}^{d}a^{(j)}(x_t^{(j)}-\tilde{x}_t^{(j)})^2\right\},
\end{aligned}
$$

completing the proof.

To get the bound for any $q \geq 1$, first note that when $q \in [1, \infty)$, the function $h : [0, \infty) \to \mathbb{R}$ given by $h(x) := x^{1/q}$ (with $h(0) := 0$) is concave for all $x > 0$. Thus, by Jensen's inequality, $\mathbb{E}\{h(X)\} \leq h(\mathbb{E}\{X\})$ for any non-negative random variable $X$. Next, we let the $q$-norm in question be given by (13), with $a \in \sigma(\hat{\mathcal{H}}_t)$ denoting the associated weighting vector, and continue as in the case of $q = 2$ above:

$$
\begin{aligned}
\mathbb{E}\left\{\frac{1}{2c_t}\|\beta_t \odot (x_t - \tilde{x}_t)\|^2\right\} &= \mathbb{E}\left\{\mathbb{E}\left\{\frac{1}{2c_t}\|\beta_t \odot (x_t - \tilde{x}_t)\|^2\big|\hat{\mathcal{H}}_t\right\}\right\} \\
&= \mathbb{E}\left\{\frac{1}{2c_t}\mathbb{E}\left\{\left(\sum_{j=1}^{d}a^{(j)}\beta_t^{(j)}\left|x_t^{(j)}-\tilde{x}_t^{(j)}\right|^q\right)^{2/q}\bigg|\hat{\mathcal{H}}_t\right\}\right\} \\
&\leq \mathbb{E}\left\{\frac{1}{2c_t}\left(\mathbb{E}\left\{\left(\sum_{j=1}^{d}a^{(j)}\beta_t^{(j)}\left|x_t^{(j)}-\tilde{x}_t^{(j)}\right|^q\right)^2\bigg|\hat{\mathcal{H}}_t\right\}\right)^{1/q}\right\},
\end{aligned}
$$

where the last inequality follows since $\mathbb{E}\left\{h(X)|\hat{\mathcal{H}}_t\right\} \leq h\left(\mathbb{E}\left\{X|\hat{\mathcal{H}}_t\right\}\right)$ by the concavity of $h$ as argued above, where $X = \left(\sum_{j=1}^{d}a^{(j)}\beta_t^{(j)}\left|x_t^{(j)}-\tilde{x}_t^{(j)}\right|^q\right)^2$. On the other hand, since $h$ is also increasing, we can bound $h\left(\mathbb{E}\left\{X|\hat{\mathcal{H}}_t\right\}\right)$ by first upper-bounding $\mathbb{E}\left\{X|\hat{\mathcal{H}}_t\right\}$. In particular,

$$
\begin{aligned}
\mathbb{E}\left\{X|\hat{\mathcal{H}}_t\right\} &= \mathbb{E}\left\{\left(\sum_{j=1}^{d}a^{(j)}\beta_t^{(j)}\left|x_t^{(j)}-\tilde{x}_t^{(j)}\right|^q\right)^2\bigg|\hat{\mathcal{H}}_t\right\} \\
&= \mathbb{E}\left\{\sum_{j=1}^{d}a^{(j)}\beta_t^{(j)}\left|x_t^{(j)}-\tilde{x}_t^{(j)}\right|^q\left(\sum_{i=1}^{d}a^{(i)}\beta_t^{(i)}\left|x_t^{(i)}-\tilde{x}_t^{(i)}\right|^q\right)\bigg|\hat{\mathcal{H}}_t\right\} \\
&\leq \mathbb{E}\left\{\sum_{j=1}^{d}a^{(j)}\beta_t^{(j)}\left|x_t^{(j)}-\tilde{x}_t^{(j)}\right|^q\left(\sum_{i=1}^{d}a^{(i)}\left|x_t^{(i)}-\tilde{x}_t^{(i)}\right|^q\right)\bigg|\hat{\mathcal{H}}_t\right\}
\end{aligned}
$$

$$= \sum_{j=1}^{d} a^{(j)} \mathbb{E}\left\{\beta_t^{(j)} | \hat{\mathcal{H}}_t\right\} \left|x_t^{(j)} - \tilde{x}_t^{(j)}\right|^q \left(\sum_{i=1}^{d} a^{(i)} \left|x_t^{(i)} - \tilde{x}_t^{(i)}\right|^q\right)$$

$$\leq p_* \sum_{j=1}^{d} a^{(j)} \left|x_t^{(j)} - \tilde{x}_t^{(j)}\right|^q \left(\sum_{i=1}^{d} a^{(i)} \left|x_t^{(i)} - \tilde{x}_t^{(i)}\right|^q\right)$$

$$= p_* \|x_t - \tilde{x}_t\|^{2q}.$$

Thus, $h\left(\mathbb{E}\left\{X | \hat{\mathcal{H}}_t\right\}\right) \leq h\left(p_* \|x_t - \tilde{x}_t\|^{2q}\right) = p_*^{1/q} \|x_t - \tilde{x}_t\|^2$. Thus,

$$\mathbb{E}\left\{\frac{1}{2c_t} \|\beta_t \odot (x_t - \tilde{x}_t)\|^2\right\} \leq \mathbb{E}\left\{\frac{1}{2c_t} h\left(\mathbb{E}\left\{X | \hat{\mathcal{H}}_t\right\}\right)\right\}$$

$$\leq \mathbb{E}\left\{\frac{1}{2c_t} p_*^{1/q} \|x_t - \tilde{x}_t\|^2\right\},$$

completing the proof of the third bound. $\qquad\square$

Thus, controlling the regret in perturbed optimization reduces to picking a suitable norm $\|\cdot\|$ and applying Lemma 2 at each time step $t$, and then controlling the differences $x_t - \tilde{x}_t$. To that end, we use the stability of ADA-FTRL updates, that is, that the difference of two ADA-FTRL iterates is controlled by the difference in the two states of the algorithm resulting in the iterates. The following lemma provides this stability bound.

**Lemma 3.** *Let $(x_t)_{t=1}^{T}$ and $(\tilde{x}_t)_{t=1}^{T}$ be given by updates* (5) *and* (6), *respectively, and suppose that Assumption 2 holds. Define $\Delta_t = r_{0:t-1}(x_t) - r_{0:t-1}(\tilde{x}_t) + \hat{r}_{0:t-1}(\tilde{x}_t) - \hat{r}_{0:t-1}(x_t)$ for $t = 1, 2, \ldots, T$. Then, for all $t = 1, 2, \ldots, T+1$,*

$$\frac{1}{2}\|x_t - \tilde{x}_t\|_{(t)}^2 \leq \frac{1}{2}\left\|\sum_{s \in O_t} I_{t,s} g_s\right\|_{(t,*)}^2 + (t - \hat{t}_t)\left(\phi(x_t) - \phi(\tilde{x}_t)\right) + \Delta_t. \tag{14}$$

*Proof of Lemma 3.* Since both $(r_t)_{t=1}^{T}$ and $(\hat{r}_t)_{t=1}^{T}$ are admissible, the ADA-FTRL margin lemma [16, Lemma 24 (Appendix F)] applied to the update (5) implies that for all $t = 1, 2, \ldots, T$,

$$\langle \hat{z}_{t-1}, \tilde{x}_t - x_t\rangle + \hat{t}_t\left(\phi(\tilde{x}_t) - \phi(x_t)\right) + \hat{r}_{0:t-1}(\tilde{x}_t) - \hat{r}_{0:t-1}(x_t) \geq \mathcal{B}_{\hat{t}_t \phi + \hat{r}_{0:t-1}}(\tilde{x}_t, x_t),$$

while for update (6) we have

$$\langle z_{t-1}, x_t - \tilde{x}_t\rangle + t\left(\phi(x_t) - \phi(\tilde{x}_t)\right) + r_{0:t-1}(x_t) - r_{0:t-1}(\tilde{x}_t) \geq \mathcal{B}_{t\phi + r_{0:t-1}}(x_t, \tilde{x}_t).$$

By the strong convexity of $t\phi + r_{0:t-1}$ and $t\phi + \hat{r}_{0:t-1}$ w.r.t. $\|\cdot\|_{(t)}$, convexity of $\phi$, and the fact that $\hat{t}_t > t > 0$ (so that $\hat{t}_t\phi + r_{0:t-1}$ is also strongly-convex w.r.t. $\|\cdot\|_{(t)}$), we have $\mathcal{B}_{\hat{t}_t\phi + \hat{r}_{0:t-1}}(\tilde{x}_t, x_t) \geq \frac{1}{2}\|x_t - \tilde{x}_t\|_{(t)}^2$ and $\mathcal{B}_{t\phi + r_{0:t-1}}(x_t, \tilde{x}_t) \geq \frac{1}{2}\|x_t - \tilde{x}_t\|_{(t)}^2$. Adding the above,

$$\frac{1}{2}\|x_t - \tilde{x}_t\|_{(t)}^2 \leq -\frac{1}{2}\|x_t - \tilde{x}_t\|_{(t)}^2 + \langle z_{t-1} - \hat{z}_{t-1}, x_t - \tilde{x}_t\rangle + (t - \hat{t}_t)\left(\phi(x_t) - \phi(\tilde{x}_t)\right)$$

$$+ \left(r_{0:t-1}(x_t) - \hat{r}_{0:t-1}(x_t)\right) - \left(r_{0:t-1}(\tilde{x}_t) - \hat{r}_{0:t-1}(\tilde{x}_t)\right)$$

$$= -\frac{1}{2}\|x_t - \tilde{x}_t\|_{(t)}^2 + \left\langle \sum_{s \in O_t} I_{t,s} g_s, x_t - \tilde{x}_t\right\rangle + (t - \hat{t}_t)\left(\phi(x_t) - \phi(\tilde{x}_t)\right) + \Delta_t$$

$$\leq \frac{1}{2}\left\|\sum_{s \in O_t} I_{t,s} g_s\right\|_{(t,*)}^2 + (t - \hat{t}_t)\left(\phi(x_t) - \phi(\tilde{x}_t)\right) + \Delta_t, \tag{15}$$

where in the last step we have used the Fenchel-Young inequality, completing the proof. $\qquad\square$

We can now prove the theorem.

*Proof of Theorem 4.* For $t = 1, 2, \ldots, T$, recall that the imaginary iterate $\tilde{x}_t$ is defined by (6)

$$\tilde{x}_t = \underset{x \in \mathcal{X}}{\arg\min} \, \langle g_{1:t-1}, x \rangle + t\phi(x) + r_{0:t-1}(x) \,,$$

and note that in addition to the difference between $r_{0:t-1}$ and $\hat{r}_{0:t-1}$, the actual iterate $x_t$ and the imaginary iterate $\tilde{x}_t$ have a difference of $\nu_t \phi(x)$ in their regularization.

Starting from the regret decomposition, and using the linear regret of the imaginary iterate $\tilde{x}_t$, as well as the fact that $r_t$ are non-negative by Assumption 2, we have

$$R_T^{(f+\phi)}(x^*) \leq \sum_{t=1}^{T} \langle g_t, \tilde{x}_t - x^* \rangle + \tilde{\epsilon}_{1:T} + \delta_{1:T} - B_{1:T} + \sum_{t=1}^{T} (\phi(\tilde{x}_t) - \phi(x^*))$$

$$\leq r_{0:T}(x^*) - \sum_{t=0}^{T} r_t(\tilde{x}_{t+1}) + \sum_{t=1}^{T} \frac{1}{2} \|g_t\|_{(t,*)}^2 + \tilde{\epsilon}_{1:T} + \delta_{1:T} - B_{1:T}$$

$$\leq r_{0:T}(x^*) + \sum_{t=1}^{T} \frac{1}{2} \|g_t\|_{(t,*)}^2 + \tilde{\epsilon}_{1:T} + \delta_{1:T} - B_{1:T} \,. \tag{16}$$

In the above, the first inequality follows by Lemma 1. The second inequality follows by bounding the linear regret $\sum_{t=1}^{T} \langle g_t, \tilde{x}_t - x^* \rangle$ using Theorem 1, and the third by dropping the non-negative terms $r_t(\tilde{x}_{t+1})$.

Next, we bound the penalty terms $\tilde{\epsilon}_{1:T}$. For each $t = 1, 2, \ldots, T$, using the fact that $\nu_t > 0$ by Assumption 2 and $\nu_t \in \sigma(\hat{\mathcal{H}}_t)$ by definition, we have

$$\mathbb{E}\{\tilde{\epsilon}_t + \phi(\tilde{x}_t) - \phi(x_t)\} \leq \mathbb{E}\left\{ \frac{p_* \nu_t}{2} \|g_t\|_{(t,*)}^2 + \frac{1}{2\nu_t} \|(x_t - \tilde{x}_t)\|_{(t)}^2 \right\}$$

$$\leq \mathbb{E}\left\{ \frac{p_* \nu_t}{2} \|g_t\|_{(t,*)}^2 \right\}$$

$$+ \mathbb{E}\left\{ \frac{1}{2\nu_t} \left( \left\| \sum_{s \in O_t} I_{t,s} g_s \right\|_{(t,*)}^2 + 2(\nu_t \phi(\tilde{x}_t) - \nu_t \phi(x_t) + \Delta_t) \right) \right\}$$

$$\leq \mathbb{E}\left\{ \frac{p_* \nu_t}{2} \|g_t\|_{(t,*)}^2 + \sum_{s \in O_t} \frac{\tau_t}{2\nu_t} \|I_{t,s} g_s\|_{(t,*)}^2 + \frac{\Delta_t}{\nu_t} + \phi(\tilde{x}_t) - \phi(x_t) \right\}$$

$$\leq \mathbb{E}\left\{ \frac{p_* \nu_t}{2} \|g_t\|_{(t,*)}^2 + \sum_{s \in O_t} \frac{\tau_t}{2\nu_t} \|g_s\|_{(t,*)}^2 + \frac{\Delta_t}{\nu_t} + \phi(\tilde{x}_t) - \phi(x_t) \right\}$$

$$\leq \mathbb{E}\left\{ \frac{p_* \nu_t}{2} \|g_t\|_{(t,*)}^2 + \sum_{s \in O_t} \frac{\tau_t}{2\nu_t} \|g_s\|_{(s,*)}^2 + \frac{\Delta_t}{\nu_t} + \phi(\tilde{x}_t) - \phi(x_t) \right\} \,. \tag{17}$$

The first inequality above uses Lemma 2 with $c_t = p_* \nu_t$ (using the assumption of $\| \cdot \|_{(t)}$ being a weighted $q$-norm with $q = 1$ or $q = 2$), the second follows by Lemma 3, the third uses the convexity of the norms $\| \cdot \|_{(t,*)}^2$ and Jensen's inequality, the forth follows because $I_{t,s}$ is a $\{0,1\}$-valued diagonal matrix and $\| \cdot \|_{(t)}$ is a weighted $q$-norm, and hence $\|I_{t,s} g_s\|_{(t,*)} \leq \|g_s\|_{(t,*)}$, and the last line follows because $s \in O_t$ implies $s \leq t$ by construction, and for $s \leq t$, the dual norms satisfy $\| \cdot \|_{(t,*)} \leq \| \cdot \|_{(s,*)}$ by Assumption 2. Summing the second term on the r.h.s. of (17), for $t = 1, 2, \ldots, T$, we get

$$\sum_{t=1}^{T} \sum_{s \in O_t} \frac{\tau_t}{2\nu_t} \|g_s\|_{(s,*)}^2 = \sum_{s=1}^{T} \left( \sum_{t:s \in O_t} \frac{\tau_t}{2\nu_t} \right) \|g_s\|_{(s,*)}^2 \,, \tag{18}$$

Thus, summing (17) over $t$, combining with (18), and noting that the terms $\phi(x_t) - \phi(\tilde{x}_t)$ cancel from the sides of the asyncrony penalty bounds (17), we get

$$\mathbb{E}\left\{ R_T^{(f+\phi)}(x^*) \right\} \leq \mathbb{E}\left\{ r_{0:T}(x^*) + \sum_{t=1}^{T} \frac{1 + p_* \nu_t + \sum_{s:t \in O_s} \frac{\tau_s}{\nu_s}}{2} \|g_t\|_{(t,*)}^2 + \frac{\Delta_t}{\nu_t} + \delta_{1:T} \right\} \,.$$

Finally, noting that $x_t \in \sigma(\mathcal{H}_t)$ by definition, by Assumption 3 it follows that $\mathbb{E}\{\delta_t | \mathcal{H}_t\} = 0$ in the stochastic setting. This completes the proof. $\qquad\square$

## B  ASYNCADA($\rho$): ASYNCADA with inexact clock

In this section, we present ASYNCADA($\rho$), a more general version of ASYNCADA that maintains the global clock sparsely. In the context of ASYNCADA($\rho$), we use $t'$ to denote the *clock* variable in the shared memory, and use $t$ to denote the virtual iteration index as we specify below. The processes run copies of Algorithm 3 concurrently. Each process is also equipped with an internal counter $t''$ and a function `MaintainClock` to control the updating of the global clock $t'$.

Similar notes as in ASYNCADA apply regarding the maintenance of the step-size $\eta$ and the formation of the average iterate. Note, however, that unlike ASYNCADA, step-sizes changing with time need to use $\hat{t}_t$ rather than $t$, since the latter is not available anymore. As Theorem 5 in Appendix C shows, this has a negligible effect on the convergence guarantees.

---

**Algorithm 3:** ASYNCADA($\rho$): ASYNCADA with inexact clock

**Input:** clock update frequency $\rho$
1  Initialize internal local counter $t'' \leftarrow 0$
2  **repeat**
3  $\quad$ $\hat{\eta} \leftarrow$ a full (lock-free) read of the shared step-sizes $\eta$
4  $\quad$ $\hat{z} \leftarrow$ a full (lock-free) read of the shared dual vector $z$
5  $\quad$ $\hat{t} \leftarrow$ `MaintainClock()` $\qquad\qquad$ // $t \leftarrow t+1$, denote $\hat{z}_{t-1} = \hat{z}$, $\hat{\eta}_t = \hat{\eta}, \hat{t}_t = \hat{t}$
6  $\quad$ Receive $\xi_t$
7  $\quad$ Compute the next iterate: $x_t \leftarrow \mathbf{prox}(\hat{t}_t\phi, -\hat{z}_{t-1}, \hat{\eta}_t)$ $\qquad$ // prox defined in (3)
8  $\quad$ Obtain the noisy gradient estimate: $g_t \leftarrow g_t(x_t, \xi_t)$
9  $\quad$ **for** $j$ such that $g_t^{(j)} \neq 0$ **do** $z^{(j)} \leftarrow z^{(j)} + g_t^{(j)}$ $\qquad$ // atomic update
10 $\quad$ Update the shared step-size vector $\eta$
11 **until** terminated

---

**Algorithm 4:** Maintaining the local and global iteration counters

1  **Function** `MaintainClock()`
2  $\quad$ Let $\gamma > t''\tau_*$
3  $\quad$ $t'' \leftarrow t'' + 1$ $\qquad\qquad\qquad$ // count number of iterations by this process
4  $\quad$ **if** $t'' \geq \rho$ **then** $\qquad\qquad$ // Update global clock every $\rho$ local iterations
5  $\quad\quad$ $t'' \leftarrow 0$
6  $\quad\quad$ $t' \leftarrow t' + \rho$ $\qquad\qquad\qquad$ // atomic read-increment
7  $\quad$ **end if**
8  $\quad$ **return** $t' + \gamma$ $\quad$ // Use the value of $t'$ read in Line 6 if executed; otherwise read $t'$
9  **end**

---

**Indexing the iterates**  Unlike ASYNCADA, in ASYNCADA($\rho$) the iterates are not physically indexed by the global clock $t$. As such, at each point in time we define a virtual count of the number of iterations undertaken so far, and then come up with an actual estimate of this virtual global clock. To that end, we use the "after-read" iteration indexing proposed by Leblond et al. [17]: we define the $t$-th iteration to be the one corresponding to the $t$-th completion of reading the shared memory (which happens by reading $t'$ in Line 6 or 8 of `MaintainClock`), before the execution of Line 6. This ensures that $\hat{z}_{t-1}$ contains only updates made by processes $s < t$, which proves useful in the analysis.

**Estimating the clock.**  In ASYNCADA($\rho$), the processes share share an integer $t'$ to estimate the (virtual) iteration count $t$, which is updated by each process every $\rho$ iterations. In particular, in each iteration a process makes one call to the function `MaintainClock` (Algorithm 4), which increments its local counter $t''$ of the number of updates made by that process since it last updated the global clock; then, after every $\rho$ local updates, `MaintainClock` increments the shared global clock estimate $t'$ by $\rho$ (and resets $t''$ for that process). Note that in this way, $t'$ is always an under-estimate of $t$, and

ASYNCADA (Algorithm 1) is recovered when $\rho = 1$. Again, when $\phi \equiv 0$ and $t$ is not used for setting the step-sizes $\eta$ either, there is no need to maintain $t'$ physically, and the call to `MaintainClock` can be ommitted in Algorithm 3.

## C  Proofs for ASYNCADA and ASYNCADA($\rho$)

We start the analysis by a lemma on the time estimates formed by ASYNCADA($\rho$).

**Lemma 4** (Time estimate of ASYNCADA($\rho$)). *Suppose ASYNCADA($\rho$) is run for $T$ iterations with any $\rho \geq 1$, using $\gamma \geq \rho\tau_* + \tau_*^2$. Then, the estimated clock $\hat{t}_t$ is non-decreasing with $t$, i.e., for all $s, t \in [T]$ with $s < t$, we have $\hat{t}_s \leq \hat{t}_t$. In addition, for all $t \in [T]$, we have $\hat{t}_t > t + \tau_*^2$.*

*Proof.* Fix $s < t \in [T]$, and note that the value of $t'$ read in `MaintainClock` in the $s$-th iteration cannot be greater than the value of $t'$ read in the the $t$-th iteration. Specifically, $t'$ can only increase over (physical) time, and the iterations are indexed by the time they make their last reading of the shared memory before the update in Line 7 of ASYNCADA($\rho$), which is the reading (and possibly incrementing) of $t'$ in Line 8 (respectively, Line 6) of `MaintainClock`. Thus, the reading of $t'$ in iteration $s < t$ necessarily has happened before that of $t$, leading to a smaller value of $t'$. As all the processes are adding the same fixed value of $\gamma$ to $t'$ to obtain $\hat{t}$, this implies $\hat{t}_s \leq \hat{t}_t$.

To see that $\hat{t}_t > t$, fix $t$ and let $t'$ be the value of the global clock estimate at the end of the call to `MaintainClock` in the $t$-th iteration. Since there have been at most $\rho - 1$ updates in each of the $\tau$ processors since the last update of $t'$ by each processor, $t' > t - \rho\tau \geq t - \rho\tau_*$, where the second inequality holds by the definition of $\tau_*$. As such, $\hat{t}_t = t' + \gamma \geq t' + \rho\tau_* + \tau_*^2 > t + \tau_*^2$. $\qquad\square$

*Proof of Theorem 2.* The theorem follows immediately from the convergence bound of ASYNCADA($\rho$) with $\rho = 1$, given by Theorem 5 below. $\qquad\square$

To analyze ASYNCADA($\rho$), we need to make a slightly modified version of Assumption 1, since $\hat{t}_t$ is now a non-deterministic part of the state of the algorithm:

**Assumption 4** (Independence of $\xi_t$). For all $t = 1, 2, \ldots, T$, the $t$-th sample $\xi_t$ is independent of the history $\hat{\mathcal{H}}_t = \left\{ \left( \xi_s, \hat{z}_s, \hat{t}_s, \hat{\eta}_{s+1} \right)_{s=1}^{t-1} \right\}$.

**Theorem 5.** *Suppose that either all $f_t, t \in [T]$ are convex, or $\phi \equiv 0$ and $f_t \equiv f$ for some star-convex function $f$. Consider ASYNCADA($\rho$) running under Assumption 4 for $T > \tau_*^2$ updates, using $\gamma = 2\tau_*^2$ and any $\rho \leq \tau_*$ in `MaintainClock`. Let $\eta_0 > 0$. Then:*

(i) *If $\mathbb{E}\{\|g_t\|_2^2\} \leq G_*^2$ for all $t \in [T]$, then using a fixed $\eta_t = \eta_0\sqrt{T}$ or a time-varying $\eta_t = \eta_0\sqrt{\hat{t}_t}$,*

$$\frac{1}{T}\mathbb{E}\left\{ R_T^{(f+\phi)} \right\} \leq \frac{1}{\sqrt{T}} \left( \eta_0\|x^*\|_2^2 + \frac{2(1 + p_*\tau_*^2)}{\eta_0}G_*^2 \right). \tag{19}$$

(ii) *If for all $\xi \in \Xi$, $F(\cdot, \xi)$ is convex and 1-smooth w.r.t. a norm $\|\cdot\|_l$, then given a constant $c_0 > 8(1 + p_*\tau_*^2)$ and using a fixed $\eta_{t,i} = c_0 l_i + \eta_0\sqrt{T}$ or a time-varying $\eta_{t,i} = c_0 l_i + \eta_0\sqrt{\hat{t}_t}$,*

$$\frac{1}{T}\mathbb{E}\left\{ R_T^{(f+\phi)} \right\} \leq \frac{c_0\|x^*\|_l^2}{T} + \frac{2}{\sqrt{T}} \left( \eta_0\|x^*\|_2^2 + \frac{4(1 + p_*\tau_*^2)}{\eta_0}\sigma_*^2 \right), \tag{20}$$

*where $\sigma_*^2 = \mathbb{E}\{\|g(x^*, \cdot)\|_2^2\}$.*

(iii) *If $\phi$ is $\mu$-strongly-convex and $\mathbb{E}\{\|g_t\|_2^2\} \leq G_*^2$ for all $t \in [T]$, then using $\eta_t \equiv 0$ or, equivalently, $\mathbf{prox}(\hat{t}_t\phi, -z, 0) := \arg\min_{x \in \mathcal{X}} \hat{t}_t\phi(x) + \langle z, x \rangle = \nabla\phi^*(-z/\hat{t}_t)$,*

$$\frac{1}{T}\mathbb{E}\left\{ R_T^{(f+\phi)} \right\} \leq \frac{(1 + p_*\tau_*^2)G_*^2(1 + \log(T))}{\mu T}, \tag{21}$$

*Proof.* We cast ASYNCADA($\rho$) in the Perturbed-ADA-FTRL framework of Section 4:

(i) Thanks to the after-read time-indexing discussed above, $\hat{z}_t$ in ASYNCADA($\rho$) cannot include any coordinate updates from $g_s$ for $s > t$ since by construction, the reading of $z$ in $t$ has finished before calculating of $g_s$ is started. As such, $\hat{z}_{t-1}$ and $z_{t-1}$ are related to each other by (11) for all $j \in [d]$ and $t \in [T]$.

(ii) In addition, letting $r_{0:t-1} = \hat{r}_{0:t-1} = \frac{1}{2}\|\cdot\|_{\eta_t}$, it is easy to see that the ASYNCADA($\rho$) update $x_t \leftarrow \mathbf{prox}(\hat{t}_t\phi, -\hat{z}_{t-1}, \hat{\eta}_t)$ is equivalent to the perturbed ADA-FTRL update (5).

(iii) Furthermore, by Lemma 4, $\hat{t}_t$ is non-decreasing with, and greater than, $t$. Thus, $\eta_t$ is also non-decreasing with $t$, and $\hat{t}_t$, $r_{0:t}$ and $\hat{r}_{0:t}$ satisfy Assumption 2 with norms $\|\cdot\|_t = \|\cdot\|_{\eta_t}$.

(iv) Finally, Assumption 4 ensures that Assumption 3 holds.

Therefore, letting $\nu_t = \hat{t}_t - t$, applying Theorem 4, and noting that $\Delta_t = 0$ by construction, we get

$$\mathbb{E}\left\{R_T^{(f+\phi)}(x^*)\right\} \leq \mathbb{E}\left\{r_{0:T}(x^*) + \sum_{t=1}^{T} \frac{1 + p_*\nu_t + \sum_{s:t\in O_s}\frac{\tau_s}{\nu_s}}{2}\|g_t\|_{(t,*)}^2 - B_{1:T}\right\}.$$

Next, the assumption that either $f_t$ is convex, or $f_t \equiv f$ for a star-convex $f$ implies $\mathcal{B}_f(x^*, x_t) \geq 0$; hence, the $B_{1:T}$ terms can be dropped. Also, by construction, $\hat{t}_t \leq t + \gamma$, as $t'$ always under-estimates $t$. Together with Lemma 4 and since $\gamma = 2\tau_*^2$, this implies $\tau_*^2 < \nu_t \leq 2\tau_*^2$. Furthermore, $\tau_s$ is the number of processes iteration $s$ is in read-conflict with, while $\{s : t \in O_s\}$ is the set of iterations $t$ is in write-conflict with; hence $\tau_s \leq \tau_*$ and $|\{s : t \in O_s\}| \leq \tau_*$, implying that the summation above is bounded by 1 and

$$\mathbb{E}\left\{R_T^{(f+\phi)}(x^*)\right\} \leq \mathbb{E}\left\{r_{0:T}(x^*) + \sum_{t=1}^{T} \frac{1 + 2p_*\tau_*^2 + 1}{2}\|g_t\|_{(t,*)}^2\right\}. \tag{22}$$

Using the definition of $G_*$, the fact that $r_{0:T} = \frac{\sqrt{\hat{t}_T}}{2}\|\cdot\|^2 \leq \frac{\sqrt{T+\gamma}}{2}\|\cdot\|^2 \leq \frac{\sqrt{3T}}{2}\|\cdot\|^2$, the expansion $\|\cdot\|_{(t,*)}^2 = \frac{1}{\eta_t}\|\cdot\|^2$, and the well-known bound $\sum_{t=1}^{T}(\sqrt{t})^{-1} \leq 2\sqrt{T}$ [21], we get (19).

To get (20), we continue from (22) but instead upper-bound $\|g_t\|$ as follows:

$$\frac{1}{2}\|g_t\|_{(t,*)}^2 \leq \|\nabla F(x_t, \xi_t) - \nabla F(x^*, \xi_t)\|_{(t,*)}^2 + \|\nabla F(x^*, \xi_t)\|_{(t,*)}^2$$

$$\leq \frac{1}{c_0}\|\nabla F(x_t, \xi_t) - \nabla F(x^*, \xi_t)\|_{l,*}^2 + \frac{1}{\eta_0\sqrt{t}}\|\nabla F(x^*, \xi_t)\|^2,$$

where the last step follows by the definition of $\eta_{t,i}$, which in particular implies $\|\cdot\|_{(t)}^2 \geq c_0\|\cdot\|_l^2$ and $\|\cdot\|_{(t)}^2 \geq \eta_0\sqrt{\hat{t}_t}\|\cdot\|^2 \geq \eta_0\sqrt{t}\|\cdot\|^2$ (similarly, in case of a fixed step size, $\|\cdot\|_{(t)}^2 \geq \eta_0\sqrt{T}\|\cdot\|^2 \geq \eta_0\sqrt{t}\|\cdot\|^2$). Putting back into (22), we obtain

$$\mathbb{E}\left\{R_T^{(f+\phi)}(x^*)\right\} \leq \mathbb{E}\left\{\frac{c_0}{2}\|x^*\|_l + \frac{\eta_0\sqrt{3T}}{2}\|x^*\|^2 + \sum_{t=1}^{T} \frac{2 + 2p_*\tau_*^2}{\eta_0\sqrt{t}}\|\nabla F(x^*, \xi_t)\|^2\right\}$$

$$+ \mathbb{E}\left\{\sum_{t=1}^{T} \frac{2 + 2p_*\tau_*^2}{c_0}\|\nabla F(x_t, \xi_t) - \nabla F(x^*, \xi_t)\|_{\ell,*}^2\right\}$$

$$\leq \mathbb{E}\left\{\frac{c_0}{2}\|x^*\|_l + \frac{\eta_0\sqrt{3T}}{2}\|x^*\|^2 + \sum_{t=1}^{T} \frac{2 + 2p_*\tau_*^2}{\eta_0\sqrt{t}}\sigma_*^2\right\}$$

$$+ \mathbb{E}\left\{\sum_{t=1}^{T} \frac{1}{4}\|\nabla F(x_t, \xi_t) - \nabla F(x^*, \xi_t)\|_{l,*}^2\right\}$$

$$\leq \mathbb{E}\left\{\frac{c_0}{2}\|x^*\|_l + \frac{\eta_0\sqrt{3T}}{2}\|x^*\|^2 + \frac{2(2 + 2p_*\tau_*^2)}{\eta_0}\sigma_*^2\sqrt{T}\right\}$$

$$+ \mathbb{E}\left\{ \sum_{t=1}^{T} \frac{1}{2} (F(x_t, \xi_t) - F(x^*, \xi_t) - \langle \nabla F(x^*, \xi_t), x_t - x^* \rangle) \right\}$$

$$= \mathbb{E}\left\{ \frac{c_0}{2} \|x^*\|_l + \frac{\eta_0 \sqrt{3T}}{2} \|x^*\|^2 + \frac{4(1 + p_* \tau_*^2)}{\eta_0} \sigma_*^2 \sqrt{T} \right\}$$

$$+ \mathbb{E}\left\{ \sum_{t=1}^{T} \frac{1}{2} (f(x_t) - f(x^*) - \langle \nabla f(x^*), x_t - x^* \rangle) \right\}$$

$$= \mathbb{E}\left\{ \frac{c_0}{2} \|x^*\|_l + \frac{\eta_0 \sqrt{3T}}{2} \|x^*\|^2 + \frac{4(1 + p_* \tau_*^2)}{\eta_0} \sigma_*^2 \sqrt{T} \right\}$$

$$+ \mathbb{E}\left\{ \sum_{t=1}^{T} \frac{1}{2} (f(x_t) - f(x^*) - \langle \phi'(x^*), x^* - x_t \rangle) \right\}$$

$$= \mathbb{E}\left\{ \frac{c_0}{2} \|x^*\|_l + \frac{\eta_0 \sqrt{3T}}{2} \|x^*\|^2 + \frac{4(1 + p_* \tau_*^2)}{\eta_0} \sigma_*^2 \sqrt{T} \right\}$$

$$+ \mathbb{E}\left\{ \sum_{t=1}^{T} \frac{1}{2} (f(x_t) - f(x^*) + \phi(x_t) - \phi(x^*)) \right\}.$$

Here, the first inequality follows by the definition of $\| \cdot \|_{(t,*)}$ and the fact that $\eta_T = \eta_0 \sqrt{\hat{t}_T} \leq \eta_0 \sqrt{T + \gamma} \leq \eta_0 \sqrt{3T}$, the second inequality follows by the definition of $c_0$ and $\sigma$, the third follows by smoothness of $F$ [23] and the bound $\sum_{t=1}^{T} (\sqrt{t})^{-1} \leq 2\sqrt{T}$ [21], the fourth by the independence of $\xi_t$ from the history, the fifth by the optimality of $x^*$ (where $\phi'(x^*)$ denotes the sub-gradient of $\phi$ for which $\phi'(x^*) + \nabla f(x^*) = 0$), and the last line follows by convexity of $\phi$. Moving the last term to the l.h.s. and multiplying the sides by 2 completes the proof of (20).

To prove (21), note that $t\phi$ is $t\mu$-strongly-convex by assumption, and thus the sequence of regularizers $r_t = \hat{r}_t = 0$ still satisfy Assumption 2 with the norms $\| \cdot \|_{(t,*)}^2 = \mu t \| \cdot \|^2$. Thus, (22) implies

$$\mathbb{E}\left\{ R_T^{(f+\phi)}(x^*) \right\} \leq \mathbb{E}\left\{ \sum_{t=1}^{T} \frac{2(1 + p_* \tau_*^2)}{2\mu t} \|g_t\|^2 \right\}$$

$$\leq \frac{2(1 + p_* \tau_*^2)}{2\mu} G_*^2 (1 + \log(T)),$$

where in the last step we have used the bound $\sum_{t=1}^{T} (1/t) \leq 1 + \log(T)$, completing the proof. $\quad\square$

# D  Analysis of ASYNCADA($\rho$) with ADAGRAD Step-Sizes

In this section, we provide the details of the analysis of ASYNCADA($\rho$) with the step-size vector $\hat{\eta}$ tuned adaptively in parallel (e.g., similar to ASYNC-ADAGRAD of Duchi et al. [10]), but with the full generality of ASYNCADA($\rho$), including with the use of composite objectives $\phi$ and an estimated global clock.

First, recall from Section 5 that in this case, Line 10 is replaced with Line 10*: similarly to $z$, a vector is maintained in the shared memory, storing the sum of squares of the observed gradient values for each coordinate individually. Also, this vector is read and updated in the same way as $z$ (with a square-root applied to each coordinate to recover the vector $\hat{\eta}$ before passing it to the **prox** operator in Line 7). Finally, note that similarly to $\hat{z}_{t-1}$, $\eta_t^{(j)}$ will lack some of the updates from the iterations happening concurrently with the $t$-th iteration. The after-read time-indexing defined in Appendix B ensures that these concurrent updates have indices less than $t$, that is, there exists a set $D_{t,j} \subset [t-1]$ such that for $s \in D_{t,j}$, $\alpha^2 \left( g_s^{(j)} \right)^2$ has not yet been added to the shared memory when $\hat{\eta}_t^{(j)}$ is read.

Formally, for all $j \in [d]$ and all $t \in [T]$, we have

$$\hat{\eta}_t^{(j)} = \alpha \sqrt{\lambda_j + \sum_{s=1}^{t-1} \left(g_s^{(j)}\right)^2 - \sum_{s \in D_{t,j}} \left(g_s^{(j)}\right)^2}, \tag{23}$$

for some hyper-parameter $\lambda_j > 0$ that, as usual in the dual-averaging formulations of ADAGRAD (cf. McMahan [21]), ensures the initial $\hat{\eta}_1$ is well-defined. Then, we have the following theorem:

**Theorem 6.** *Suppose that either all $f_t, t \in [T]$ are convex, or $\phi \equiv 0$ and $f_t \equiv f$ for some star-convex function $f$. Consider* ASYNCADA($\rho$) *running under Assumption 4 for $T \geq 1$ updates, using $\gamma = 2\tau_*^2$ and any $\rho \leq \tau_*$ in* MaintainClock. *For all $j \in [d]$, assume that $\left|g_t^{(j)}\right| \leq G_j, t \in [T]$, and $|x^{(j)}| \leq R$ for all $x \in \mathcal{X}$. Then, using* ADAGRAD *step-sizes given by (23) with $\lambda_j = (\tau_* + 1)G_j^2$ and any $\alpha > 0$, we have*

$$\mathbb{E}\left\{R_T^{(f+\phi)}\right\} \leq \frac{\alpha R^2 \sqrt{\tau_* + 1}}{2} \sum_{j=1}^d G_j + \left(\alpha R^2 + \frac{2 + 2p_*\tau_*^2}{\alpha}\right) \sqrt{T} \sum_{j=1}^d G_j \sqrt{p_j}. \tag{24}$$

**Remark 5.** For a fixed $\tau_*$, the first term in (24) is of lower order. Thus, if $p_*\tau_*^2 \leq c$ for a constant $c$ and we tune $\alpha$ based on $R$, Theorem 6 implies that the average iterate $\bar{x}_T$ of ASYNCADA($\rho$) with ADAGRAD step-sizes satisfies

$$\mathbb{E}\{f(\bar{x}_T) - f(x^*)\} = \mathcal{O}\left(R \sum_{j=1}^d G_j \sqrt{p_j/T}\right),$$

which, up to a constant factor, is the same worst-case convergence rate as that of serial composite-objective ADAGRAD under sparsity (see, e.g., Duchi et al. [9, Corollary 1]). In addition, when $\phi = 0$, this is the best rate attainable by any serial algorithm under sparsity [10, Proposition 1].[10] Thus, Theorem 6 generalizes the linear speed-up result of ASYNC-ADAGRAD to non-box-shaped $\mathcal{X}$, star-convex objectives, and proximal updates using a convex $\phi$ with an inexact global clock. It should be noted, however, that the speed-up regime of ASYNC-ADAGRAD with box-shaped constraints is less restrictive than that of Theorem 6, i.e., the condition required by Duchi et al. [10, Eq. (8)] is $p_*\tau_* \leq c$ rather than $p_*\tau_*^2 \leq c$.

*Proof.* The proof follows as in the case of Theorem 5, in particular noting the implications of the after-read indexing, the fact that Assumption 4 implies Assumption 3, and the equivalence of the **prox** operator to the Perturbed-ADA-FTRL update (5) given a Euclidean regularizer. However, here we use different $r_{0:t}$ and $\hat{r}_{0:t}$, and hence $\Delta_t$ does not necessarily vanish anymore. In particular, we let $\tilde{\eta}_t$ to be given by

$$\tilde{\eta}_t^{(j)} = \alpha \sqrt{\lambda_j + \sum_{s=1}^{t-1} \left(g_s^{(j)}\right)^2}$$

and let the idealized $r_{0:t-1}$ be given by

$$r_{0:t-1}(x) = \frac{1}{2}\|x\|_{\tilde{\eta}_t}^2 = \sum_{j=1}^d \frac{\alpha}{2} \sqrt{\lambda_j + \sum_{s=1}^{t-1} \left(g_s^{(j)}\right)^2} \left(x^{(j)}\right)^2,$$

whereas the actual regularizer used by ASYNCADA($\rho$) is given by (23), i.e.,

$$\hat{r}_{0:t-1}(x) = \frac{1}{2}\|x\|_{\hat{\eta}_t}^2 = \sum_{j=1}^d \frac{\alpha}{2} \sqrt{\lambda_j + \sum_{s=1}^{t-1} \left(g_s^{(j)}\right)^2 - \sum_{s \in D_{t,j}} \left(g_s^{(j)}\right)^2} \left(x^{(j)}\right)^2.$$

$$f(x) - f(x^*) \leq \langle \nabla f(x), x - x^* \rangle = \mathbb{E}\{\langle \nabla F(x, \xi), x - x^* \rangle\} \leq \sum_{j \in [d]} G_j p_j R.$$

Note that for all $t \in [T]$, this ensures that $\tilde{\eta}_t^{(j)} \geq \hat{\eta}_t^{(j)} \geq \eta_t^{(j)} := \alpha \sqrt{\sum_{s=1}^{t} \left( g_s^{(j)} \right)^2}$ for all $j \in [d]$, since by assumption, $\lambda_j = (\tau_* + 1) G_j^2 \geq \sum_{s \in D_{t,j} \cup \{t\}} \left( g_t^{(j)} \right)^2 \geq \sum_{s=t-\tau_*}^{t} \left( g_t^{(j)} \right)^2$. Hence, both $r_{0:t-1}$ and $\hat{r}_{0:t-1}$ are strongly-convex w.r.t. the norm $\frac{1}{2} \| \cdot \|_{\eta_t}$, and these norms are non-decreasing (since $\eta_t$ is non-decreasing by definition). Combined with Lemma 4 and the fact that $r_t \geq 0$ as defined above, this implies that Assumption 2 is satisfied. Hence, we can apply Theorem 4 with $\nu_t = \hat{t}_t - t$, recalling that by assumption of (star-)convexity of $f_t$, we have $B_{1:T} \geq 0$, and obtain

$$
\begin{aligned}
\mathbb{E}\left\{ R_T^{(f+\phi)} \right\} &\leq \mathbb{E}\left\{ \frac{1}{2} \|x^*\|_{\tilde{\eta}_T}^2 + \sum_{t=1}^{T} \left( \frac{1 + p_* \nu_t + \sum_{s:t \in O_s} \frac{\tau_s}{\nu_s}}{2} \|g_t\|_{(t,*)}^2 + \frac{\Delta_t}{\nu_t} \right) \right\} \\
&\leq \mathbb{E}\left\{ \frac{1}{2} \|x^*\|_{\tilde{\eta}_T}^2 + \sum_{t=1}^{T} \left( \frac{2 + 2 p_* \tau_*^2}{2} \|g_t\|_{(t,*)}^2 + \frac{\Delta_t}{\nu_t} \right) \right\} \\
&\leq \mathbb{E}\left\{ \sum_{j=1}^{d} \frac{\alpha R^2}{2} \sqrt{\lambda_j + \sum_{t=1}^{T} \left( g_t^{(j)} \right)^2} + \sum_{t=1}^{T} \frac{\Delta_t}{\nu_t} \right\} \\
&\quad + \mathbb{E}\left\{ (2 + 2 p_* \tau_*^2) \sum_{t=1}^{T} \sum_{j=1}^{d} \frac{1}{2\alpha \sqrt{\sum_{s=1}^{t} \left( g_t^{(j)} \right)^2}} \left( g_t^{(j)} \right)^2 \right\} \\
&\leq \mathbb{E}\left\{ \sum_{j=1}^{d} \frac{\alpha R^2}{2} \sqrt{\lambda_j + \sum_{t=1}^{T} \left( g_t^{(j)} \right)^2} + \sum_{t=1}^{T} \frac{\Delta_t}{\nu_t} \right\} \\
&\quad + \mathbb{E}\left\{ \frac{2 + 2 p_* \tau_*^2}{\alpha} \sum_{j=1}^{d} \sqrt{\sum_{t=1}^{T} \left( g_t^{(j)} \right)^2} \right\} \\
&\leq \frac{\alpha R^2}{2} \sum_{j=1}^{d} \sqrt{\lambda_j} + \sum_{j=1}^{d} \frac{\alpha R^2}{2} \sqrt{\sum_{t=1}^{T} \mathbb{E}\left\{ \left( g_t^{(j)} \right)^2 \right\}} + \sum_{t=1}^{T} \mathbb{E}\left\{ \frac{\Delta_t}{\nu_t} \right\} \\
&\quad + \frac{2 + 2 p_* \tau_*^2}{\alpha} \sum_{j=1}^{d} \sqrt{\sum_{t=1}^{T} \mathbb{E}\left\{ \left( g_t^{(j)} \right)^2 \right\}}, \quad (25)
\end{aligned}
$$

where the second line follows as (22) in the proof of Theorem 5, the third line uses the definition of $\tilde{\eta}_T$ and $\eta_t$ and the upper-bound $R$, the fourth line uses the ADAGRAD lemma (McMahan [21, Lemma 4]), and the last line uses concavity of square-root and Jensen's inequality.

Next, we bound the terms $\Delta_t$. Recall that by definition, $\tilde{\eta}_t^{(j)} \geq \hat{\eta}_t^{(j)}$. Hence, $r_{0:t-1} \geq \hat{r}_{0:t-1}$, and

$$
\begin{aligned}
\Delta_t &= r_{0:t-1}(x_t) - r_{0:t-1}(\tilde{x}_t) + \hat{r}_{0:t-1}(\tilde{x}_t) - \hat{r}_{0:t-1}(x_t) \leq r_{0:t-1}(x_t) - \hat{r}_{0:t-1}(x_t) \\
&= \sum_{j=1}^{d} \frac{\tilde{\eta}_t^{(j)} - \hat{\eta}_t^{(j)}}{2} \left( x_t^{(j)} \right)^2 \\
&\leq \frac{\alpha R^2}{2} \sum_{j=1}^{d} \left( \sqrt{\lambda_j + \sum_{s=1}^{t-1} \left( g_s^{(j)} \right)^2} - \sqrt{\lambda_j + \sum_{s=1}^{t-1} \left( g_s^{(j)} \right)^2 - \sum_{s \in D_{t,j}} \left( g_s^{(j)} \right)^2} \right) \\
&\leq \sum_{j=1}^{d} \frac{\alpha R^2 \sum_{s \in D_{t,j}} \left( g_s^{(j)} \right)^2}{4 \sqrt{\lambda_j + \sum_{s=1}^{t-1} \left( g_s^{(j)} \right)^2 - \sum_{s \in D_{t,j}} \left( g_s^{(j)} \right)^2}}
\end{aligned}
$$

$$\leq \sum_{j=1}^{d} \frac{\alpha R^2 \sum_{s \in D_{t,j}} \left(g_s^{(j)}\right)^2}{4\sqrt{\sum_{s=1}^{t} \left(g_s^{(j)}\right)^2}} \, ,$$

where the third line uses the upper-bound $R$ on $\left|x_t^{(j)}\right|$, the fourth line uses the inequality $\sqrt{a+b} - \sqrt{a} \leq \frac{b}{2\sqrt{a}}$ which holds for all $a, b > 0$, and the last line uses $\eta_t^{(j)} \leq \hat{\eta}_t^{(j)}$. Noting that by construction, either $\tau_* \geq 1$ (and thus, $\tau_* \leq \tau_*^2 < \nu_t$) or $\Delta_t = 0$, we have

$$\sum_{t=1}^{T} \frac{\Delta_t}{\nu_t} \leq \frac{\alpha R^2}{4} \sum_{j=1}^{d} \sum_{t=1}^{T} \sum_{s \in D_{t,j}} \frac{\left(g_s^{(j)}\right)^2}{\tau_* \sqrt{\sum_{k=1}^{t} \left(g_k^{(j)}\right)^2}}$$

$$\leq \frac{\alpha R^2}{4} \sum_{j=1}^{d} \sum_{t=1}^{T} \sum_{s \in D_{t,j}} \frac{\left(g_s^{(j)}\right)^2}{\tau_* \sqrt{\sum_{k=1}^{s} \left(g_k^{(j)}\right)^2}}$$

$$\leq \frac{\alpha R^2}{4} \sum_{j=1}^{d} \sum_{s=1}^{T} \sum_{t:s \in D_{t,j}} \frac{\left(g_s^{(j)}\right)^2}{\tau_* \sqrt{\sum_{k=1}^{s} \left(g_k^{(j)}\right)^2}}$$

$$\leq \frac{\alpha R^2}{2} \sum_{j=1}^{d} \sum_{s=1}^{T} \frac{\left(g_s^{(j)}\right)^2}{2\sqrt{\sum_{k=1}^{s} \left(g_k^{(j)}\right)^2}}$$

$$\leq \frac{\alpha R^2}{2} \sum_{j=1}^{d} \sqrt{\sum_{t=1}^{T} \left(g_t^{(j)}\right)^2} \, ,$$

where the second line uses the fact that $s < t$, the third line swaps the summations on $t$ and $s$, the fourth line uses the fact that $|\{t : s \in D_{t,j}\}| \leq \tau_*$, and the fifth line uses the ADAGRAD lemma (McMahan [21, Lemma 4]). Putting back into (25) and using the concavity of square-root with Jensen's inequality again, we obtain

$$\mathbb{E}\left\{R_T^{(f+\phi)}\right\} \leq \frac{\alpha R^2}{2} \sum_{j=1}^{d} \sqrt{\lambda_j} + \sum_{j=1}^{d} \alpha R^2 \sqrt{\sum_{t=1}^{T} \mathbb{E}\left\{\left(g_t^{(j)}\right)^2\right\}}$$

$$+ \frac{2 + 2p_* \tau_*^2}{\alpha} \sum_{j=1}^{d} \sqrt{\sum_{t=1}^{T} \mathbb{E}\left\{\left(g_t^{(j)}\right)^2\right\}} \, .$$

Using the fact that $\mathbb{E}\left\{\left(g_t^{(j)}\right)^2\right\} \leq p_{t,j} G_j^2$, the definition of $\lambda_j$, and simplifying, we obtain

$$\mathbb{E}\left\{R_T^{(f+\phi)}\right\} \leq \frac{\alpha R^2 \sqrt{\tau_* + 1}}{2} \sum_{j=1}^{d} G_j + \left(\alpha R^2 + \frac{2 + 2p_* \tau_*^2}{\alpha}\right) \sqrt{T} \sum_{j=1}^{d} G_j \sqrt{p_j} \, ,$$

This completes the proof. $\qquad\qquad\square$

## E  Proofs for HEDGEHOG

*Proof of Theorem 3.* As in the case of ASYNCADA($\rho$), the proof follows by casting HEDGEHOG as Perturbed-ADA-FTRL. In particular, the same relation between $\hat{z}_{t-1}$ and $z_{t-1}$ holds, and it is easy to see that with the after-read time-indexing the HEDGEHOG update corresponds to Perturbed-ADA-FTRL with the regularizer $r_{0:t}(x) = r(x) + \eta \ln(d)$ where $r(x) = \eta \sum_{i=1}^{d} x^{(i)} \log\left(x^{(i)}\right)$, which

is 1-strongly-convex w.r.t. the $\ell_1$ norm [34]. Note also that we can assume any value for $\hat{t}_t > t$, including $\hat{t}_t = t + \nu_t$ for any $\nu_t > 0$, as we don't use $\hat{t}_t$ in the update and hence don't need to be able to compute it. Then, Assumption 2 is satisfied with $\|\cdot\|_t = \sum_{i=1}^{d} |x^{(i)}|$ being the $\ell_1$ norm, with $\|\cdot\|_{(t,*)} = \|\cdot\|_\infty$. Then, applying Theorem 4 and noting $\phi = 0, \Delta_t = 0, B_{1:T} \geq 0$, we have

$$\mathbb{E}\left\{R_T^{(f)}(x^*)\right\} \leq r(x^*) + \eta \log(d) + \sum_{t=1}^{T} \mathbb{E}\left\{\frac{1 + p_*\nu_t + \sum_{s:t\in O_s} \frac{\tau_s}{\nu_s}}{2}\|g_t\|_{(t,*)}^2\right\},$$

for any $\nu_t$ determined by $\hat{\mathcal{H}}_t$. In particular, letting $\nu_t = \tau_*/\sqrt{p_*}$, recalling that $\tau_s$ and $|\{s : t \in O_s\}|$ cannot be larger than $\tau_*$, and noting that $r(x^*) \leq 0$ for any $x^* \in \mathcal{X}$ completes the proof. $\qquad\square$

# F    Extra details for the analysis of serial ADA-FTRL

A typical proxy for bounding the regret of serial optimization algorithms is *linearizing* the loss, and studying the linearized regret [5, 34, 14]. In particular, we define the linearized forward regret

$$R_T^+(x^*) = \sum_{t=1}^{T} \langle g_t, x_{t+1} - x^* \rangle,$$

and carry out the analysis in two steps: a decomposition of the regret in terms of the forward regret $R_T^+$, followed by a bound on $R_T^+$. To that end, we need the following assumption.

**Definition 2** (Admissible regularizers.). A sequence of regularizer functions $(r_t)_{t=0}^{T}$ is "admissible" for ADA-FTRL if and only if all $r_t$ are defined on a common convex domain $S \subset \mathbb{R}^d$, the intersection $\mathcal{X} \cap S$ is non-empty, and there exists a sequence of norms $\left(\|\cdot\|_{(t)}\right)_{t=1}^{T}$ such that for all $t = 1, 2, \ldots, T$, the cumulative regularizer $t\phi + r_{0:t-1} : S \to \mathbb{R}$ is lower-semi-continuous and 1-strongly-convex w.r.t. $\|\cdot\|_{(t)}$.

As shown by Lemma 5, admissible regularizers guarantee that the ADA-FTRL updates (2) are well-defined, that is, there exists some $x_{t+1} \in \mathcal{X}$ that satisfies (2), and the associated optimal value is finite.

**Lemma 5** (Well-posed ADA-FTRL). *For all $t = 0, 1, \ldots, T$, the argmin sets that define $x_{t+1}$ in the* ADA-FTRL *updates (2) are non-empty, and their optimal values are finite.*

*Proof.* Fix $t \in [T]$, and consider the extended-value function $h_t = \langle z_{t-1}, \cdot \rangle + r_{0:t-1} + \mathcal{I}_{x \in S \cap \mathcal{X}}$, which is proper, l.s.c. and convex by construction. In addition, since $r_{0:t-1}$ is strongly-convex over $S$, then $h_t$ is l.s.c. and 1-strong-convex on $\mathbb{R}^d$. The result then follows by Proposition 17.26 of [1], noting that $x_t$ will be the corresponding minimizer by definition. $\qquad\square$

Then, we have the following bound on the regret of ADA-FTRL.

**Theorem 7** (Forward regret of ADA-FTRL, [16].). *For any $x^* \in \mathcal{X}$ and for any sequence of linear losses $\langle g_t, \cdot \rangle, t = 1, 2, \ldots, T$, and using any sequence of admissible regularizers $r_0, r_1, \ldots, r_T$, the forward regret of* ADA-FTRL *satisfies*

$$R_T^+(x^*) \leq r_{0:T}(x^*) - \sum_{t=0}^{T} r_t(x_{t+1}) + \sum_{t=1}^{T} (\phi(x^*) - \phi(x_t)) - \sum_{t=1}^{T} \mathcal{B}_{r_{0:t-1}}(x_{t+1}, x_t). \qquad (26)$$

Theorem 1 follows as a direct consequence of the above theorem by using the strong convexity of $r_{0:t-1}$ and the Fenchel-Young inequality; see [16] for further details.