[Reviews · NeurIPS 2019]

Reviewer 1



Summary This paper concerns the asynchronous sparse online and stochastic optimization settings. In this setting several algorithms work in parallel to optimize the same objective. The difficulty herein lies that not all algorithms are aware of the current state of the objective, complicating the analysis. Existing convergence guarantees in this setting only hold for box shaped constraint sets. In this paper the authors develop several new algorithms that can deal with non-box shaped constrained sets: “AsynCADA” and “HedgeHog!”. Both these algorithms are able to deal with inexact readings of the current time point in addition to dealing with the previously mentioned difficulties. Both these algorithms are analysed in a setting related to online convex optimization in which the algorithms only have access to a perturbed state. This setting allows for the use of standard online convex optimization algorithms, but now the analysis requires us the bound the perturbation penalty between the actual update and an update in which the true state is known. Qualitative assessment. The build-up of the paper could be improved. While the description of the problems, algorithms, and analysis was fine as is I would prefer that the paper was reordered. The way the paper is set up now is that first the algorithms are presented and then the perturbed ADA-FTRL framework is introduced. I would present it the other way around, first the framework then the algorithms. This allows the reader to understand some of the design choices of the algorithms by relating it to the analysis of the framework. It even makes sense in the appendix where first the proofs of both AyncADA and HedgeHOG are presented and then the analysis of the framework is presented. However, the proofs of both algorithms hinge on the analysis of framework, which explains some of the choices made in the design of the algorithms. The new algorithms seem to be a useful contribution to the asynchronous composite optimization setting since these algorithms allow for more general constraint sets than in previous work. The generality of the analysis is a plus, as several variants of the assumptions on the loss functions are presented which allows the reader to understand the framework. The analysis itself was interesting and I agree with the authors when they say that the framework could be of independent interest. Minor comments Line 32: sothcastic gradient → stochastic gradient Line 93: f() → f_t() Line 123: prox_\phi(z, \eta) → prox(\phi, z, \eta) to be consistent with (2) Line 136: an over-estimates → an over-estimate The definitions in line 524 in the appendix and (9) of the imaginary iterate are not the same, I think that (t+1) should be t in line 524 Post-rebuttal update The authors addressed my concerns about the ordering of the paper. Therefore, I have increased my score for this paper.

Reviewer 2



In my opinion, the main advantage over existing results such as [Recht et al, Hogwild: A Lock-Free Approach to Parallelizing Stochastic Gradient Descent] and [Duchi et al, Estimation, Optimization, and Parallelism when Data is Sparse] is the type of performance bound. While the bounds in the cited papers are mainly concerned with sparsity of the data, the bound in Theorem 1 is a generic regret bound of the order O(\sqrt{T}) for the convex case and O(\log T) for the strongly convex case. This is clearly related to delayed feedback results in prediction with expert advice. The asymptotic behaviour of the regret bound for HedgeHog! is also O(\sqrt{T}). There is also an advantage in the required conditions. The paper considers the case of convex (rather than strictly convex) target function in combination with a generic convex (rather than box-shaped) domain. All these conditions are well-known in the theory of gradient decent algorithms (and discussed at length in [Bottou et al, Optimization Methods for Large-Scale Machine Learning]), but not in the theory of asynchronous algorithms. The result of [Recht et al, Hogwild: A Lock-Free Approach to Parallelizing Stochastic Gradient Descent] on asynchronous algorithms is limited to strictly convex functions and the result of [Duchi et al, Estimation, Optimization, and Parallelism when Data is Sparse] to box-shaped domains.

Reviewer 3



Post feedback: 1. thanks, this clarifies the speed up confusion. So, you want the factor p^*\tau^* to be not too large, both by ensuring high sparsity and by having a small enough number of parallel processors that \tau^* does not blow up. 2. Got it, but this does limit the usability of the algorithm to a narrower class of problems that adhere to p^*<1. I don't recall too many practical examples of this in standard machine / deep learning. 3. I think you may be missing a \mu in your expression, it should be f+\phi - \frac {\mu} 2 ||x||^2 ? I see what you are saying here though. Thanks for the clarification; I'm happy to raise my evaluation to 7. --------------- Originality: The authors claim a broad-framework style contribution that subsumes a lot of open or previously solved analysis cases. Clarity: The paper, while written fluently, is addressed to an audience that is well-versed with the precursor work. Key terms such as "linear speedup" seems hard to grasp for me, not being from the asynchronous optimization area (see below in "improvements"). Quality& Significance: The results sound significant in the breadth of cases covered in the analysis framework proposed by the authors. A main claim here is that the analysis allows for optimization over arbitrary convex constraint sets, while previous async optim algos required unconstrained or box-constraints only. This improvement is achieved by pushing the constraints into the prox-operator (pg 4) which is assumed to be easily solvable.

[Author Response · NeurIPS 2019]

We would like to thank all reviewers for their comments and questions.

**Reviewer 2:** We appreciate your recommendation about reordering the paper. We chose the current ordering because
we thought that a detailed discussion of the framework early on may make the paper somewhat less accessible to
researchers not directly familiar with the main challenges in the design and analysis of asynchronous algorithms. Yet,
we agree that the current ordering does not fully distill the view that a more experienced reader may appreciate. Thus, in
the revised version, we will present the crux of the generic framework (e.g., Section 5, which should still be accessible
to the general reader) early on, before Sections 3 & 4, and add further remarks about the design choices as appropriate.

**Reviewer 3:** Thanks for your positive comment. It is indeed correct that the regret bounds imply delayed-feedback
results for online learning. In particular, unlike the asynchronous setting, in delayed-feedback online learning (and
parameter-server models) one can typically assume that $\tau_t$ is known at time $t$. Then, using Theorem 5 with $\nu_t = \tau_t$, the
penalty term $p_* \nu_t + \sum_{s \in O_t} \frac{\tau_s}{\nu_s}$ is bounded by $\tau_t + \tau'_t$, even when $p_* = 1$. The resulting regret bound extends the delay-
dependent adaptive results in the literature (e.g., Ref [14] given in the paper, or [Joulani et al, "Delay-tolerant online
convex optimization: Unified analysis and adaptive-gradient algorithms", AAAI-2016]), and this close connection
makes it possible to extend their delay-adaptive tuning techniques to a larger set of problems.

**Reviewer 4:** Thanks for looking into the details of our paper. We have answered your questions below:

**Linear speed-up.** We agree that the definition of obtaining a linear speed-up here is somewhat specific to the
community; we are here using the methodology established in previous work (e.g., [Recht et al, "Hogwild: A Lock-Free
Approach to Parallelizing Stochastic Gradient Descent", 2011]). We have recalled the basics of the definition on Lines
31-38 of the paper, but we will expand the discussion in the final version of the paper to make it more clear.

To illustrate the definition, we first re-call the generic parallel computing definition: given a serial algorithm that solves
a given problem, a parallelized version of the algorithm is said to achieve "linear speed-up" if it solves the same problem
$(c \times P)$-times faster (in wall-clock time) than the serial algorithm, where $c \le 1$ is a constant and $P$ is the number of
processors used. For example, in a so-called "embarrasingly parallel" problem (like hyper-parameter search) where the
problem can be split into smaller, independent sub-problems, a parallel algorithm can achieve a linear speed up with
$c \approx 1$ by simply assigning different sub-problems to the $P$ processors and solving them simultaneously.

In case of a parallel asynchronous optimization algorithm, the key point is that linear speed-ups may not be possible in
general, except in specific "regimes" when the problem is, e.g., sparse enough, and the number of processors is not
too large. Theorems 1 and 2 in particular quantify this for ASYNCADA and HEDGEHOG: if for some $c'$, we have
$\tau_*^2 \le c'/p_*$, then to achieve the same accuracy as a serial algorithm would achieve after $T$ updates, ASYNCADA needs
to make $(1 + c')T$ updates (since accuracy $\approx R_T/T$, c.f. lines 104-111 of the paper). Since ASYNCADA makes
updates $P$ times faster (in wall-clock time, by doing $P$ updates in parallel), this implies that ASYNCADA solves the
same problem (achieves the same accuracy) $P/(1 + c')$ times faster than the serial algorithm would, hence enjoying a
"linear speed-up" with $c = 1/(1 + c')$ under the sparse-data regime. Importantly, the speed-up regime $p_* \tau_*^2 = O(1)$ is
the same as what was established by the previous work of Refs [10, 11, 18] of the paper.

**Definition of $p_*$.** Your example is correct. In such a case, the problem is simply not sparse, since the gradient
coordinates are not zero with positive probability, and our result does not lead to a linear speed-up (neither do the results
from previous work, with the exception of [10,11] for box-shaped, unbounded, non-proximal SGD/SAGA, with general
linear speed-ups without sparsity remaining an open problem [18]). However, our regret bounds still hold, in terms of
the observed delays $\tau_t, \tau'_t$, even when $p_* = 1$, and generalize the delay-adaptive regret bounds of online optimization
(as referenced in response to Reviewer 3) to inconsistent delays and coordinate perturbations.

**Remark 3.** We will rephrase this remark to make it more clear. The aim of the remark is to compare part (iii)
of Theorem 1 to the work of Nguyen et al. [25]. Nguyen et al. [25] recognize the fact that for a strongly-convex
objective on an unbounded domain, one cannot assume that the gradients are uniformly bounded by a constant $G_*$ (the
"bounded-gradient" assumption). Hence, they provide an analysis of strongly-convex optimization using unconstrained
Hogwild! with a global clock, without relying on the bounded-gradient assumption. The aim of Remark 3 is to
emphasize that the $G_*$ bound in part (iii) of Theorem 1 is not a global upper-bound on the gradients of a strongly-convex
function: it is a global upper-bound only on the non-strongly-convex part of the function, and thus compatible with
strong convexity. Hence, like Nguyen et al. [25], Theorem 1 (iii) avoids the incompatible bounded-gradients assumption
when the objective is strongly-convex, but further applies to any constraint set without requiring a global clock.

**Typos and errors.** We will fix all the mistakes on Pages 1 and 2 and perform a further proofreading of the paper, as
well as the definition of $\hat{t}$ on Line 5 of Alg 1. Thanks for the catch!

[Meta-Review · NeurIPS 2019]

The theory of serial composite minimization for a variety of loss functions and regularizers is available. However, the same cannot be said of the theory of parallel minimization in the "lock-free" or "asynchronous" setting popularized by the Hogwild! paper of Recht et al. This paper advances the state of the art in asynchronous parallel minimization of composite convex functions (one of which is a differentiable loss and the other is a possibly non-differentiable regularizer). The out of the "box" in the title denotes the important feature of their algorithm ASYNCADA that it does not require the constraint set to be a "box" (cartesian product of intervals). They also provide an algorithm HEDGEHOG that is an asynchronous version of exponentiated gradient for optimization over the simplex. The reviewers felt that this work, while not particularly groundbreaking, is an important step forward in an area of current interest in ML. The authors did promise a number of changes that the reviewers agreed will be important for them to make in the final version. The authors should remember to make the promised changes.